# Interplay of coprecipitation and adsorption processes: deciphering amorphous mineral-organic associations under both forest and cropland conditions

Floriane Jamoteau[1,2,3,4*], Emmanuel Doelsch[2,3], Nithavong Cam[1], Clément Levard[1], Thierry Woignier[5,6], Adrien Boulineau[7], Francois Saint-Antonin[7], Sufal Swaraj[8], Ghislain Gassier[1], Adrien Duvivier[1], Daniel Borschneck[1], Marie-Laure Pons[1], Perrine Chaurand[1], Vladimir Vidal[1], Nicolas Brouilly[9], Isabelle Basile-Doelsch[1]

[1]Aix Marseille Université, CNRS, IRD, INRAE, Coll France, CEREGE, Aix-en-Provence, 13545, France
[2]CIRAD, UPR Recyclage et risque, Montpellier, F-34398, France
[3]Recyclage et Risque, Univ Montpellier, CIRAD, Montpellier, F-34398, France
[4]Institute of Earth Surface Dynamics, University of Lausanne, Lausanne, 1015, CH
[5]Campus Agro Environnemental Caraïbes-IMBE-CNRS, B.P. 214, Petit Morne, Le Lamentin, Martinique, 97232, France
[6]Laboratoire Charles Coulomb UMR 5221 CNRS-UM2, Université Montpellier 2, Montpellier Cedex5, 34095, France
[7]Université Grenoble Alpes, CEA, LITEN, Grenoble, 38100, France
[8]Synchrotron SOLEIL, L'Orme des Merisiers, Departementale 128, Saint-Aubin, 91190, France
[9]Aix-Marseille Université, CNRS UMR 7288, IBDM, Marseille, 13000, France

*Correspondence to*: Floriane Jamoteau (floriane.jamoteau@unil.ch; jamoteau@cerege.fr)

**Abstract.** Mineral-organic associations are crucial carbon and nutrient reservoirs in soils. However, conversion from forest to agricultural systems disrupts these associations, leading to carbon loss and reduced soil fertility in croplands. Identifying the types of mineral-organic associations within a single soil is already challenging, and detecting those susceptible to disruption during forest-to-crop conversion is even more complex. Yet, addressing this identification challenge is essential for devising strategies to preserve organic matter in croplands. Here, we aimed to identify the predominant mineral-organic associations within an Andosol (developed on Fe-poor parent material) under both forest and cropland conditions. To achieve this, we collected Andosol samples from both a forested and a cultivated area, located 300 meters apart. We then analysed differences between both soil profiles in soil physicochemical parameters and characterized mineral-organic associations using an array of spectro-microscopic techniques for comprehensive structural and compositional analysis. At the micro and nanoscale spatial resolution, we observed mineral-organic associations in the form of amorphous coprecipitates, composed of a mix of C+Al+Si and C+Al+Fe+Si nanoCLICs, proto-imogolites and organic matter, some Fe nanophases associated with organic matter and some metal-organic complexes. This challenges prior conceptions of mineral-organic associations in Andosols by demonstrating the presence of amorphous coprecipitates rather than solely organic matter associated with short-range order minerals (i.e. imogolite and allophanes). Moreover, chemical mappings suggested that these amorphous coprecipitates may adhere to mineral surfaces (i.e. phyllosilicates and imogolites), revealing secondary interactions of mineral-organic associations in soils. While the presence of similar amorphous coprecipitates in both the forest and crop Andosols was confirmed, the crop soil had 75 % less C in mineral-organic associations (in the 0-30 cm depth). Although the sample size for comparing land-use types is limited, these results suggest that the nature of mineral-organic associations remains identical

despite quantitative differences. This study highlights the crucial role of amorphous coprecipitates in C stabilization in Andosols and also suggests their vulnerability to disruption after 30 years of a forest-to-crop conversion, thereby challenging our understanding of the persistence of mineral-organic associations in Andosols.

**Keywords:** Mineral-organic associations, cropland, forest-to-crop conversion, Andosol, nanoscale, transmission electron microscopy, STXM

## 1 Introduction

Carbon sequestration in terrestrial ecosystems is facilitated through protecting organic compounds within mineral-organic associations from microbial access (Cotrufo et al., 2019; Lugato et al., 2021). Beyond their contribution to carbon sequestration, these associations serve as crucial nutrient reserves for plants and soil microorganisms, thereby enhancing soil fertility, which is essential for maintaining agricultural productivity (Bernard et al., 2022; Fontaine et al., 2024). However, soil cultivation disrupts these mineral-organic associations, leading to significant loss of C—a phenomenon known as 'C destabilization' (Sanderman et al., 2017; Bailey et al., 2019). In order to maintain agricultural productivity in cultivated soils, it is essential to preserve mineral-organic associations in croplands.

Mineral-organic associations have traditionally been regarded as resulting from organic matter adsorption onto mineral surfaces or coprecipitation of organic compounds with weathered elements from minerals (Basile-Doelsch et al., 2015). In soil with neutral to acidic soil pH, the predominant mineral-organic associations can include organic matter associated to short-range order minerals (e.g. ferrihydrite, imogolites, allophanes, Figure 1) or metal-organic complexes (Wagai and Mayer, 2007; Kleber et al., 2015; Chen et al., 2014; Rasmussen et al., 2018; Basile-Doelsch et al., 2020; Kleber et al., 2021). Yet, the exact mineral composition of mineral-organic associations is difficult to achieve and largely stems from indirect measurements of minerals rather than direct characterizations of the entire mineral-organic assemblage (e.g. using selective extractions; Rennert and Lenhardt, 2024). Recent advances in nanoscale (spectro)microscopy (e.g., TEM-EDX, STXM) have facilitated precise analyses of mineral-organic associations' composition and structure (Kinyangi et al., 2006; Solomon et al., 2007; Wan et al., 2007; Solomon et al., 2012; Asano et al., 2018), offering deeper insights into mineral-organic associations composition. In Andosols, i.e. soils with high concentrations of mineral-organic associations, microscopy and spectroscopy analyses raised questions about the stabilizing role of short-range order minerals in the form of imogolite or allophane for C (Levard et al., 2012). Instead, organic C was primarily associated in the form of nanosized coprecipitates of inorganic oligomers with organics (nanoCLICs). In such structures, organic molecules are linked to a few atoms of Al, Fe, or Si without crystalline structures (Tamrat et al., 2018, 2019; Jamoteau et al., 2023). These nanoCLICs phases do not fit into the spectra of metal-organic complexes because they are more heterogeneous in composition (Al, Fe, Si, and some Ca, Mg, K, etc.) and organic molecules are linked to metallic oligomers of approximately 2-3 atoms (Tamrat et al., 2019; Jamoteau et al., 2023). These findings challenge previous assumptions about the types of mineral-organic associations in Andosols developed from

basalt parent material. These results suggest that, in some situations, Andosol's mineral-organic associations may contain a more amorphous structure than earlier proposed models of short-range ordered minerals (Jamoteau et al., 2023). However, the presence of nanoCLICs rather than short-range order minerals with organic matter in Andosols derived from basaltic parent material, could be explained by two hypotheses: either organic matter directly associates with amorphous phases to form nanoCLICs instead of short-range order minerals, or the presence of Fe in the soil solution (derived from the weathering of basaltic parent material) prevents Al and Si from assembling into short-range order minerals like imogolite and allophane. In sum, these studies, showing various types of mineral-organic associations in different Andosols, now raise the question of their coexistence within the same soil and whether some types of associations are predominant. To determine if nanoCLICs, short-range ordered minerals with adsorbed organic matter and metal-organic complexes coexist in Andosols, or if one type is more prevalent, further nanoscale characterization of mineral-organic associations in Andosol is required.

In addition to characterizing existing mineral-organic associations, identifying the ones vulnerable to destruction or transformation during forest-to-crop transition is crucial for developing strategies to preserve organic matter in croplands. Disruption of mineral-organic associations in crop soils can be attributed to multiple factors: (i) disruption of soil aggregates, releasing entrapped mineral-organic associations (Bailey et al., 2019; Derrien et al., 2023; Even and Cotrufo, 2024); (ii) intensified root and microbial activities within cropping systems may accelerate mineral-organic associations disruption by priming effect (Keiluweit et al., 2015; Jilling et al., 2021; Fontaine et al., 2024); or (iii) shifts in soil physicochemical parameters, notably pH, which can weaken mineral-organic interactions (Newcomb et al., 2017; Bailey et al., 2019). However, regardless of these factors, the susceptibility of mineral-organic associations to disruption varies, depending on mineral crystallinity and binding strength (Li et al., 2017; Newcomb et al., 2017; Bernard et al., 2022). Consequently, some associations could be more prone to disruption than others, potentially altering the types of remaining mineral-organic associations after long-term soil cropping. Although the underlying mechanisms are not fully understood, some evidence suggests that associations with lower mineral crystallinity are particularly prone to disruption and exhibit faster turnover (Li et al., 2017; Hall et al., 2018). In Andosols, for instance, nanoCLICs-type mineral-organic associations, characterized by their amorphous mineral components composed of only a few atoms, could be particularly prone to disruption. Therefore, long-term cropping of Andosols may lead to the disruption of nanoCLICs-type associations, which would significantly alter the type of remaining mineral-organic associations.

This study aims to (i) identify the predominant mineral-organic associations within an Andosol (developed on Fe-poor parent material) under both forest and cropland conditions, and to explore possible differences in the types of mineral-organic associations, and (ii) to determine if mineral-organic associations in a forested Andosol developed on Fe-poor parent material (andesite) are similar to mineral-organic associations in a forested Andosol developed on Fe-rich parent material (basalt; from Jamoteau et al., 2023). Our working hypotheses posit that: (i) organic matter preferentially associates with amorphous mineral phases rather than short-range order minerals in relatively young Andosols (< 100 kyrs) developed on Fe-poor parent material, (ii) nanoCLICs are particularly prone to physicochemical transformations induced by cultivation, making them susceptible to destruction, and shifting the predominant mineral-organic association in the cultivated Andosol from

nanoCLICs-type to adsorption of organic matter onto short-range order minerals. To probe these hypotheses, we sampled an Andosol formed on a Fe-poor parent material (andesite parent material). We conducted analyses on two Andosol topsoil that are 300 m apart, one under forest and the other subjected to three decades of cultivation. We then identified differences between the forest and crop soil profiles in physicochemical parameters and characterized mineral-organic associations by employing an array of spectro-microscopic techniques including TEM-EDX, TEM-EELS, and STXM for comprehensive structural and compositional analysis. Although only single profiles per land use type were compared (limiting broader site-level or land-use comparisons), this approach was chosen to enable high-resolution imaging and direct visualization of mineral-organic associations.

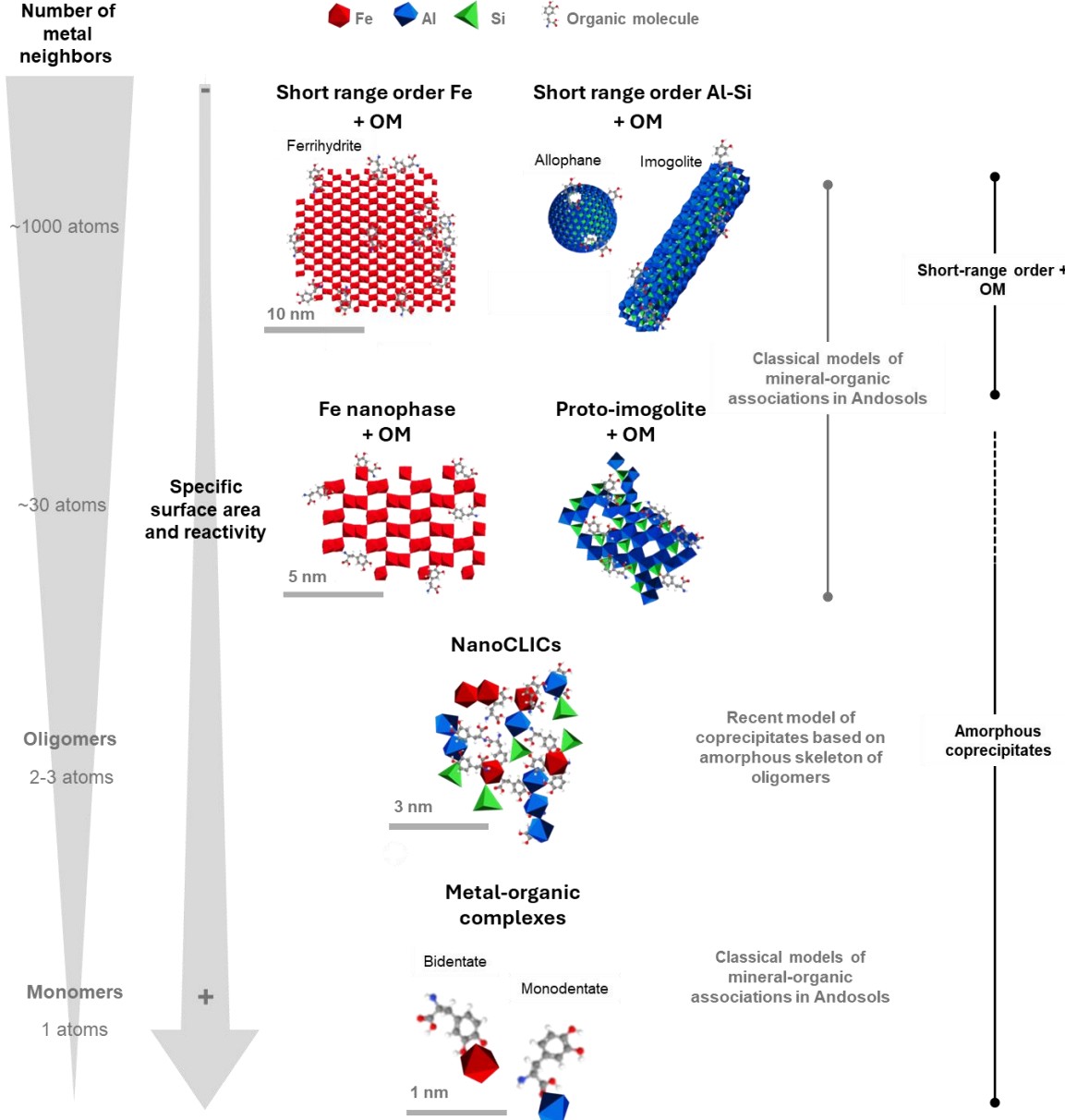

**Figure 1. Molecular models of mineral-organic associations in Andosols.** Mineral-organic associations can be described as follows: (i) an inorganic phase consisting of clusters of 100 to 1000 atoms, forming an organized crystal structure known as short-range orders minerals (SROs). These SROs include ferrihydrite (composed of Fe; representation adapted from Kleber et al. 2015) or imogolites or allophanes (composed of Si and Al; representation adapted from Levard et al. 2012); (ii) an inorganic phase with approximately 30 atoms showing limited atomic organization at local scale, such as Fe nanophases (adapted from Kleber et al 2015) and proto-imogolites (adapted from Levard et al. 2012); (iii) an inorganic phase with small amorphous oligomers formed by the polymerization of 2-3 atoms of Al, Fe and Si, without local arrangement, referred to as nanoCLICs, (adapted from Tamrat et al.(2019) and Jamoteau (2023)); and (iv) an inorganic component involving a single monomer (e.g., Fe, Al) complexed with organic molecules.

## 2 Material and methods

### 2.1 Soil sampling

Two Andosol profiles were sampled on La Martinique island (French West Indies), formed few 100 years ago (with the most recent volcanic eruption in 1929) on andesite parent material. One profile was located under a forest (14°46'31" N; 61°2'31" W), while the other was situated 300 meters away in an area converted to agriculture 30 years ago (14°46'27" N; 61°2'57" W). Both profiles were situated on a flat area at 300 meters above sea level, in a humid tropical climate (average temperature of 25°C and average annual precipitation of 3000 mm. year$^{-1}$). The crop soil, converted 30 years ago, transitioned from a forest to a banana plantation followed by a cropping system with 3-year rotations of taro, sweet potatoes, yams, and fallow periods. During the crop rotation system, ploughing was carried out to a depth of 30 to 40 cm. The differences observed between these two soil profiles may primarily result from their different land uses: one soil has remained in a forest for 30 years, while the other has been converted for agriculture. Soil sampling was carried out by opening soil pits and subsequently sampling 3 samples per soil horizon (on different sides of the pits), for a total sampling of ~1 kg of soil per horizon (i.e. at 0-5, 5-10, 10-20, 20-30, 30-40, 40-50, 50-60, 60-70, 70-80 cm). After sampling, the samples were kept humid at 4°C.

### 2.2 Monitoring differences in soil physico chemical parameters between the forest and cop soils

To monitor differences between the forest and the crop Andosols profiles, analyses on bulk soils and fine fractions (primarily composed of C in the form of mineral-organic associations) were carried out. The fine fractions from both soils, were isolated through wet sieving: briefly, 10-20 g of air-dried soil was added to 100 mL of milliQ water and sonicated (16 J.mL$^{-1}$). The 100 mL suspension was then wet sieved using a 20 µm mesh. After a minimum 12-hour settling period at 4°C of the < 20 µm fraction, the supernatant was removed, and the sedimented fraction was dried at 30°C.

The C content of bulk soils and fine fractions was analyzed using dry combustion with a Thermo FlashSmart elemental analyzer. The mineralogy of bulk soils and fine fractions was analyzed on free powder deposited on a silicon sample holder. This analysis was performed using an X-ray diffractometer (Co-Kα source at 40 mA; λ = 1.79; 2-75°; time step of 0.033°, Philips P3710 X-ray). The quantity of poorly crystalline phases in bulk soils was quantified through sequential extractions using Na-pyrophosphate, ammonium oxalate-acid, and dithionite-citrate bicarbonate (Tamm, 1922; Pansu and Gautheyrou, 2006). After extractions, solubilized Fe, Al, and Si were measured using inductively coupled plasma atomic emission spectroscopy (ICP-AES).

Soil organic carbon stocks (*SOC stock*, kg.m$^{-2}$) were calculated using the following equations (Poeplau et al., 2017; Quéro et al., 2021):

$$SOC\ stock_i = BD_i \times TOC_i \times e_i \div 10 \qquad \text{equation 1}$$

where *i* refers to the considered soil horizon, and *n* is the number of analyzed horizons. *TOC* represents the soil carbon concentration [g·kg$^{-1}$], BD is the bulk density [g·cm$^{-3}$], and e is the layer thickness [cm]. A correction was applied to compare cumulated C stocks at equivalent mass, avoiding differences in bulk density between the two sites for the same depth (Ellert

and Bettany, 1995; Poeplau et al., 2017). The reference soil mass was the heaviest horizon. This correction of the C stock (*SOC equivalent_mass*) was applied to all cumulative horizons from 0 to 80 cm, following equation (2).

$$SOC\ stock_{\text{equivalent\_mass}} = SOC\ stock_n + \left( TOC_{n+1} * \frac{M_{n\ heaviest} - M_n}{10} \right) \ (2) \qquad \text{equation 2}$$

where $n$ is the considered horizon; $SOC\ stock_n$ is the uncorrected cumulative SOC stock in kg·m$^{-2}$; $M_n$ heaviest is the heaviest soil mass [g·cm$^{-2}$] of the two sites, and $M_n$ is the lightest soil mass [g·cm$^{-2}$], and $TOC$ is the soil C concentration.

## 2.3 Characterization of mineral-organic associations

### 2.3.1 Probing types of mineral-organic associations using TEM images and TEM-EDX and TEM-EELS chemical mappings

For microscopy analyses, fine fractions of the forest and the crop soil were isolated through sedimentation: 2g of soil from the 10-20 cm horizon was added to 35 mL of milliQ water and sonicated at low power to induce minimal disaggregation (16 J.mL$^{-1}$ ; Just et al., 2021). After a minimum of 1 hour of sedimentation, the brown supernatant (with a gel-like texture) was collected and stored at 4°C. Before microscopy analysis, the suspension was diluted into ultrapure water (1/100), and 5 to 7 μL were deposited on copper grids coated with a lacey carbon film (porous film). The grids were air-dried for a few minutes before microscopy analysis. The analyses were performed using a transmission electron microscope (TEM) FEI Tecnai Osiris at 200 kV, coupled with an Energy-Dispersive X-ray spectroscopy (EDX) detection system (Super-X EDS). Imaging was conducted in bright field (direct beam) and dark field (annular dark field and high angular dark field; diffracted beam) mode. Chemical mapping was carried out using scanning transmission electron microscopy (STEM-EDX) and electron energy loss spectroscopy (TEM-EELS).

EDX Mapping was conducted with acquisition times varying from 15 to 90 minutes (with an electron dose of 100 e.Å$^{-2}$.s$^{-1}$). The chemical mapping obtained through EDX was analyzed using Esprit software (version 1.9, Bruker), and atomic proportions (at.%) were determined using the PB-ZAF algorithm. STEM-EDX mapping were conducted at scales ranging from ~100 nanometers to few micrometers. Twelve EDX mappings were carried out on the fine fractions of the forest soil, while 4 mappings were carried out on the fine fractions of the crop soil. From these mappings, various zones were selected to quantify atomic proportions, ensuring micrometric representativeness of analyzed microscopy grids (27 zones for the forest soil and 9 for the crop soil). To examine potential atomic composition heterogeneities among these zones, a principal component analysis of C, Fe, Al, and Si proportions followed by a $k$-means cluster analysis (with three imposed clusters) was performed on the selected zones using Rstudio software (using the 'stats', 'ggplot2', and 'FactoMineR' packages). Despite the lack of significant differences between the clusters, we have retained a 3 $k$-means clustering (with an imposed number of clusters) to illustrate the variability in atomic composition across areas.

To ensure elemental co-localizations of C with specific elements (Al, Si, and Fe) down to a few nanometers, TEM-EELS mappings were conducted on fine fractions of both forest and crop soils. These analyses followed protocols outlined in Jamoteau et al., (2023). Briefly, the energy range examined spanned the following edges: C K-edge (284 eV), O K-edge (532

eV), Fe L-edge (708 eV), Al K-edge (1560 eV), and Si K-edge (1839 eV). To minimize beam damage while ensuring precise elemental detection at the intended scales, the analysis time per pixel was kept as short as possible. EELS data collection occurred in two phases: initially from 250 to 1224 eV with a pixel analysis duration between 0.05 to 0.09 seconds; subsequently from 1,050 to 2,074 eV with an analysis time of 0.1-1.5 seconds per pixel. The compilation of nanoscale-resolved elemental co-localizations was executed using a Python script available in Jamoteau et al., (2023).

### 2.3.2. Probing organic matter types within mineral-organic associations using STXM

To investigate the types of organic matter within mineral-organic associations, elemental maps of C, Fe, and Al, along with speciation maps of C (K-edge), were analyzed using scanning transmission X-ray microscopy (STXM) on fine fractions of forest and crop Andosols (from the 10-20 cm horizon, see part 2.2 for details on fine fraction separation). For these mappings, the fine fractions were re-humidified with ultrapure water to form a paste consistency. This mixture was instantly frozen using liquid nitrogen, and 400-nm-thick sections were sliced at cryogenic temperature using a cryo-ultramicrotome (equipped with a diamond knife, Leica UC7). These sections were placed onto $Si_3N_4$ windows (75 nm thick, 1x1 mm, 100 μm; AGS172-3T @ Oxford Instruments) and air-dried. STXM analyses took place at the SOLEIL synchrotron (France) on HERMES beamline, where energy calibration was conducted for C (using $CO_2$ at 2 mbar) and Fe (using Fe oxides). At the edges of C, Fe, and Al, transmitted photons were recorded every 50 nm across a 5 μm x 5 μm area. The acquisition time was set to 3 ms for C and Fe edges and 5 ms for the Al edge at intervals of 50 nm. The samples were first analyzed at the C K-edge with varying scan parameters: 1 eV increments between 274 and 281 eV; 0.125 eV increments between 282.125 and 292 eV; 0.353 eV increments between 292 and 304 eV; followed by increments of 10 eV between 314 and 334 eV. Subsequently, analyses at the Fe L-edge included scan parameters of 0.5 eV increments between 700 and 705 eV; 0.15 eV increments between 705.15 and 712 eV; and finally, increments of 0.5 eV between 712.5 and 730 eV. Lastly, at the Al K-edge, increments of 0.5 eV was used between 1570 and 1600 eV. Background ($I_0$) measurements were taken concurrently with sample analysis in an adjacent area for all edges. The C, Al, and Fe maps resulted from energy subtractions at specific intervals: C at 291.35 - 275 eV; Fe at 709.3 - 700 eV; Al at 1588.8 -1580.2 eV. Spectra from these zones were normalized using Athena software (version V0.9.26; Ravel and Newville, 2005). To assess variations in C speciation within the mappings, spectral principal component analysis of the C K-edge was performed using Orange software (version 3.35.0). However, no statistically distinct clusters were identified in the results. Areas were then selected to illustrate the uniformity of C speciation across the maps.

**3 Results**

**3.1 Comparison of soil physicochemical parameters between the forest and crop soils**

To investigate the differences between the forest and crop soil profiles, and select the appropriate horizon with
quantitative differences in mineral-organic associations for micro and nanoscale mappings, we compared key physicochemical
parameters between the forest and crop Andosol profiles (Fig. 2). Results showed differences between the two soil profiles on
surface horizons only, from 0 to 30 cm depth. Firstly, compared to the forest soil, the crop soil exhibited a 46% less C stock
on the 0-40 cm depth, with a cumulative C difference of 66 kgCm$^{-2}$. Additionally, the amount of C in the fine fractions (<20
µm, Fig. 2B), attributed to C in the form of mineral-organic associations (MAOM), was lower in the crop soil compared to the
forest soil, with respectively an average of 12 and 51 gMAOM-C.kg$^{-1}$, indicating 75% less C in mineral-organic associations
in crop topsoil. In addition, differences in pH were noticed on the 0-30 cm depth, with an averaged pH of 6.3 in the forest soil
and 5.6 in the crop soil. Regarding mineralogy, both bulk and fine fractions of the forest and crop Andosols exhibited the same
mineral composition including pyroxenes, orthopyroxenes, plagioclases, titanomagnetite, quartz, and gibbsite, as well as
poorly crystalline phases (analyzed by XRD, see SI1). However, quantitative analysis of amorphous and poorly crystalline
minerals, using sequential extractions with pyrophosphate and oxalate (Rennert, 2018; Rennert and Lenhardt, 2024), showed
differences between the forest and crop soils. The amount of Al, Fe and Si extracted by pyrophosphate was twice lower in the
crop soil compared to the forest soil (on the 0-20 cm depth), indicating a lower amount of amorphous mineral phases in the
crop topsoil. On the contrary, oxalate extractions did not show a clear difference between the forest and the crop soil, indicating
similar amount of poorly-crystalline minerals in both soil profile.

In summary, the comparative analysis of soil physicochemical parameters highlighted differences between the forest
and crop soil profiles (from 0-30 cm depth), by showing a lower C stock, a 75% less C in the form of mineral-organic
associations, a lower abundance of amorphous mineral phases, and a variation in physicochemical parameters, as indicated by
a 0.7 pH disparity.

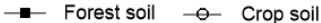

**Figure 2. Soil physicochemical parameters of the forest and crop Andosols.** (A) Cumulated C stock (corrected for equivalent mass). (B) Total C content and proportion of C in the form of particulate organic matter (POM) and mineral-associated organic matter (MAOM) along the soil profiles. (C) pH analyses along the soil profiles and (D) amount of secondary mineral phases made of amorphous phases (extracted by pyrophosphate) and poorly crystalline mineral phases (extracted by oxalate).

### 3.2 Probing types of mineral-organic associations in the forest and crop soils

### 3.2.1 Mineral-organic associations in the form of coprecipitates

To identify the types of mineral-organic associations in the forest and crop soils, we conducted chemical mappings using STEM-EDX (Fig. 3) on the fine fractions of the 10-20 cm horizons of both Andosols (selected for their differences in C content and physico-chemical parameters, Fig. 2). These analyses confirmed previous mineralogical findings (from XRD diffractograms and sequential extractions) by identifying two distinct crystallographic phases: (1) crystalline mineral phases, ranging from a few hundred nanometers to micrometers in size, typically displaying rod-like shapes, and (2) an amorphous phase, likely in contact with the rod-like minerals (Fig. 3a and Fig. S2 for extra mappings). Across all mapped mineral-organic associations in both soils, C was invariably found within the electron-amorphous phase (see Fig. 3A-B). Notably, the C was never isolated; it consistently co-occurred with a mix of Al, Si and Fe in the electron-amorphous phase (Fig. 3C). This pattern was particularly noticeable in atomic proportions along the horizontal profile that sequentially crossed crystalline minerals (on the left side of the profile in Fig. 3D) and then amorphous phases (on the right side of the profile in Fig. 3D, see additional profiles in SI2), showing a marked C proportion increase upon entering and within the amorphous phase. These profiles, together with C mapping, demonstrated that C predominantly was located in the amorphous phase, closely associated with Al (~20%), Si (~15%), and to a lesser extent Fe (~3%). This nanoscale amorphous association of C with Al, Si, and Fe demonstrated that mineral-organic associations in both Andosols were formed through the co-precipitation of elements derived from mineral weathering (mainly Al, Si and Fe) with organic matter, hereafter referred to as 'coprecipitates'. The interaction between coprecipitates and crystalline mineral phases may be secondary. While we cannot rule out the possibility that this interaction is induced by sample preparation (e.g., weak sonication followed by air-drying), the coprecipitate-mineral interaction has been found in three different mappings and even observed between mineral sheets (Figure S2), suggesting that such interactions may occur in soils.

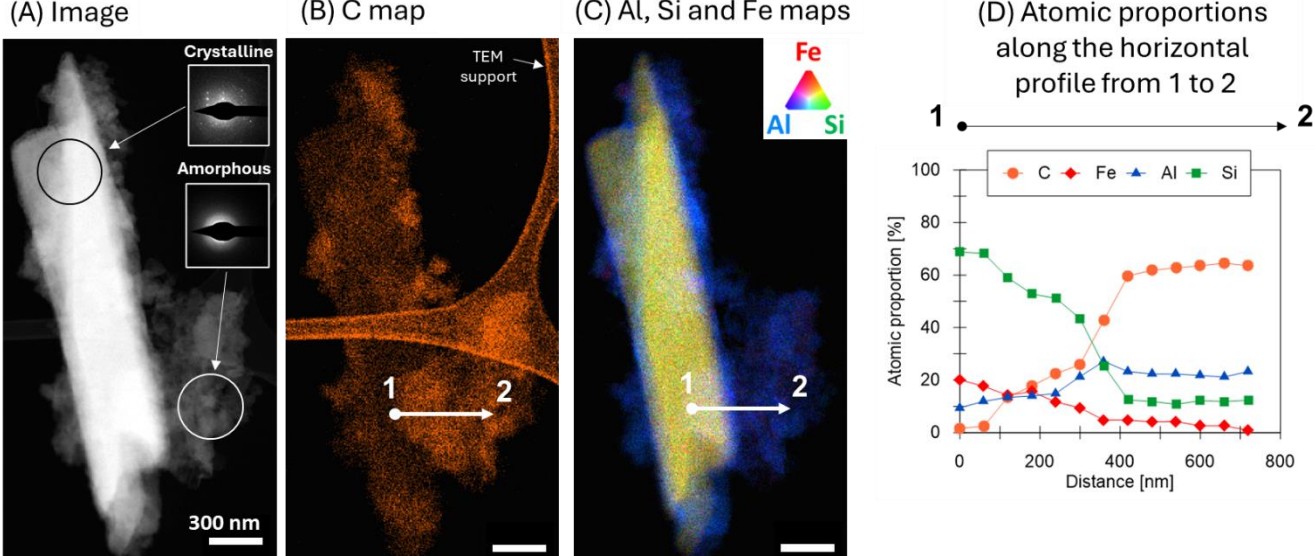

**Figure 3. Chemical mapping of mineral-organic associations using STEM-EDX.** This mapping were conducted on the fine fractions of the crop andosol. For the complete set of mineral-organic association mappings, refer to SI2. (A) TEM image in dark-field mode with electron-diffraction pattern in the boxes showing crystallinity of phases. (B) carbon mapping. (C) Al, Si, Fe mappings using RGB format; the white arrow indicates the location of the 700 nm horizontal profile shown in D. (D) atomic proportions of C, Al, Si, Fe along the horizontal profile.

### 3.2.2 Nanoscale structure and composition of coprecipitates

To determine the composition and nanoscale structure of the amorphous phase containing coprecipitates, we conducted high-resolution transmission electron microscopy (HRTEM) to image the coprecipitates from micrometer to nanometer scales. Figure 4 presents six images from the forest soil, representative of all the acquired images from both soils (see SI3 for all HRTEM images). At the micrometer to sub-micrometer scale, we primarily observed aggregates and intertwined filaments (Fig. 4A-C). These filaments are indicative of imogolites, a short-range ordered mineral forming tubular

structures of Al and Si, typically resulting from ash and pumice weathering (Wada, 1985; Wada and Harward, 1974; Parfitt, 2009; Levard and Basile-Doelsch, 2016). At the nanometer scale (Fig. 4D-F), the aggregated phase primarily appeared amorphous, except for localized crystalline planes within the aggregates (as shown in Fig. 4E). The filaments have a diameter of about few nanometers (Fig. 4F), aligning with characteristics of imogolite or bundles of imogolites (Levard and Basile Doelsch, 2016), which were very fragile under the electron beam leading to amorphization of part of bundles. These results

were observed across numerous distinct areas (see SI3) in both forest and crop soils. In sum, nanoscale images demonstrated that (1) the fine fractions were predominantly composed of amorphous phases, but (2) exhibited short-range order minerals typical of imogolite, (3) and showed local arrangement with some crystallinity (such as small Fe or Al oxyhydroxides) , which remains very minor compared to amorphous phases.

   Then, to identify associated elements with C within the amorphous phase, 12 STEM-EDX mappings were conducted

on the fine fraction from the forest soil and 4 mappings on the fine fraction from the crop soil. The results, consistent across both soils, demonstrated that C was predominantly colocalized with the amorphous phase in aggregate form (Fig. 5, see SI4 for all mappings), with minimal presence on filamentous structures associated with imogolites (indicated by arrows in Fig. 5). Further STEM-EELS mappings confirmed nanoscale colocalization of C with Al, Si, and Fe, within both forest and crop coprecipitates, below 15 nm scales (refer to SI5). These results indicated that the mineral component is primarily composed of

a mix of amorphous Al, Si, and Fe coprecipitates, even at scales down to 15 nm, and not composed of short-range ordered minerals like imogolite, allophane. The nanoscale colocalization of these elements (C, Al, Si, and Fe) demonstrates the presence of organic molecules coprecipitated with a mineral part such as inorganic oligomers, proto-imogolite or Fe nanophases resulting from parent-andesite minerals weathering. This characterization demonstrates the wide range of coprecipitates occurring at nanoscales, such as proto-imogolite+OM, nanoCLICs, metal-organic complexes and some Fe-

nanophases+OM.

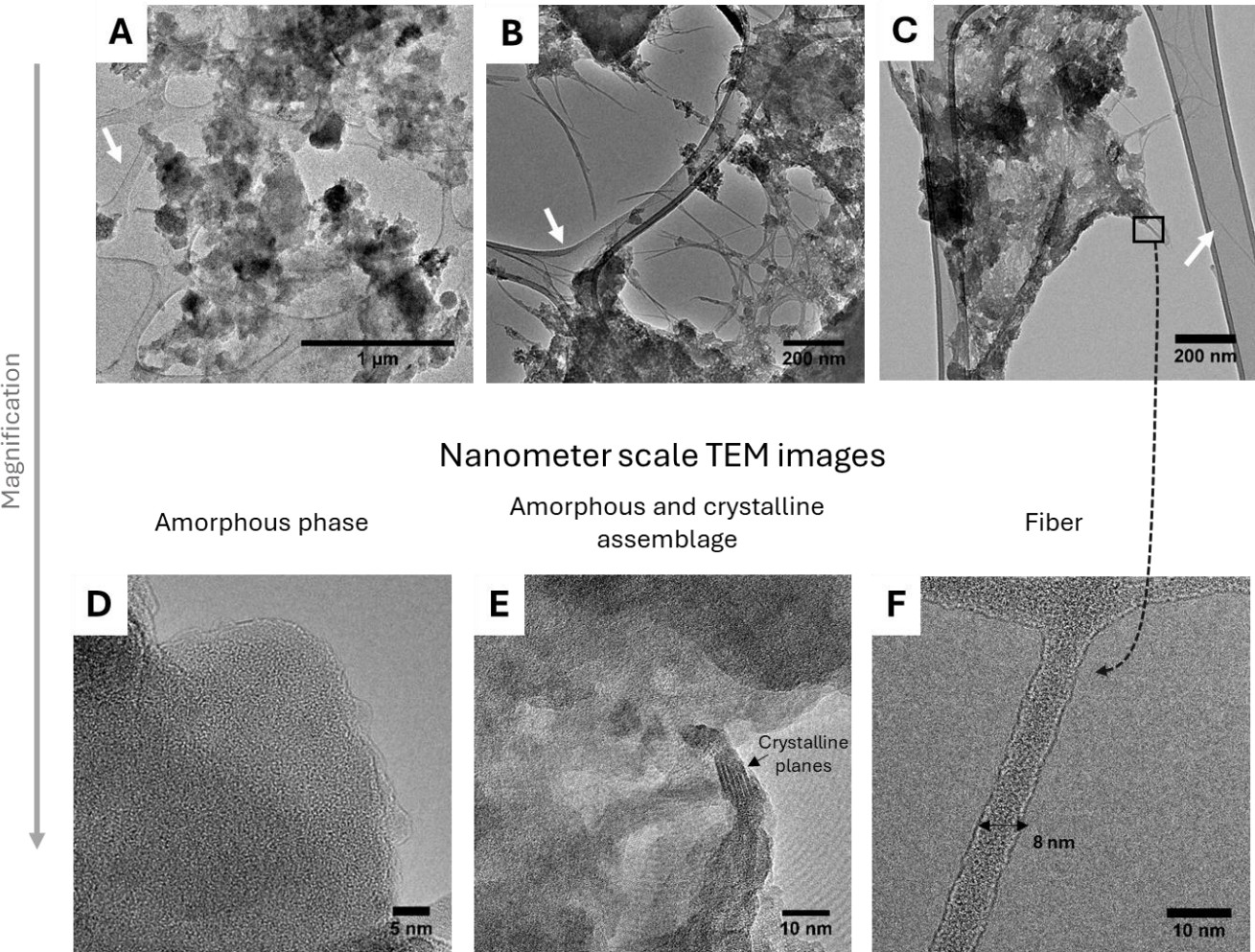

**Figure 4. High-resolution TEM analysis of the amorphous aggregated phase and filaments.** Microscale (A-C) and nanoscale (D-F) images of the amorphous aggregated phase and filaments. (D) Nanoscale image of the amorphous aggregated phase. (E) Localized crystallization within the aggregates (see crystalline planes). (F) Nanoscale image of an ~8 nm diameter filament, characteristic of imogolite minerals. The sample holder is denoted by white arrows. These images were acquired on the fine fraction of the forest soil; for additional images in both soils, refer to SI3.


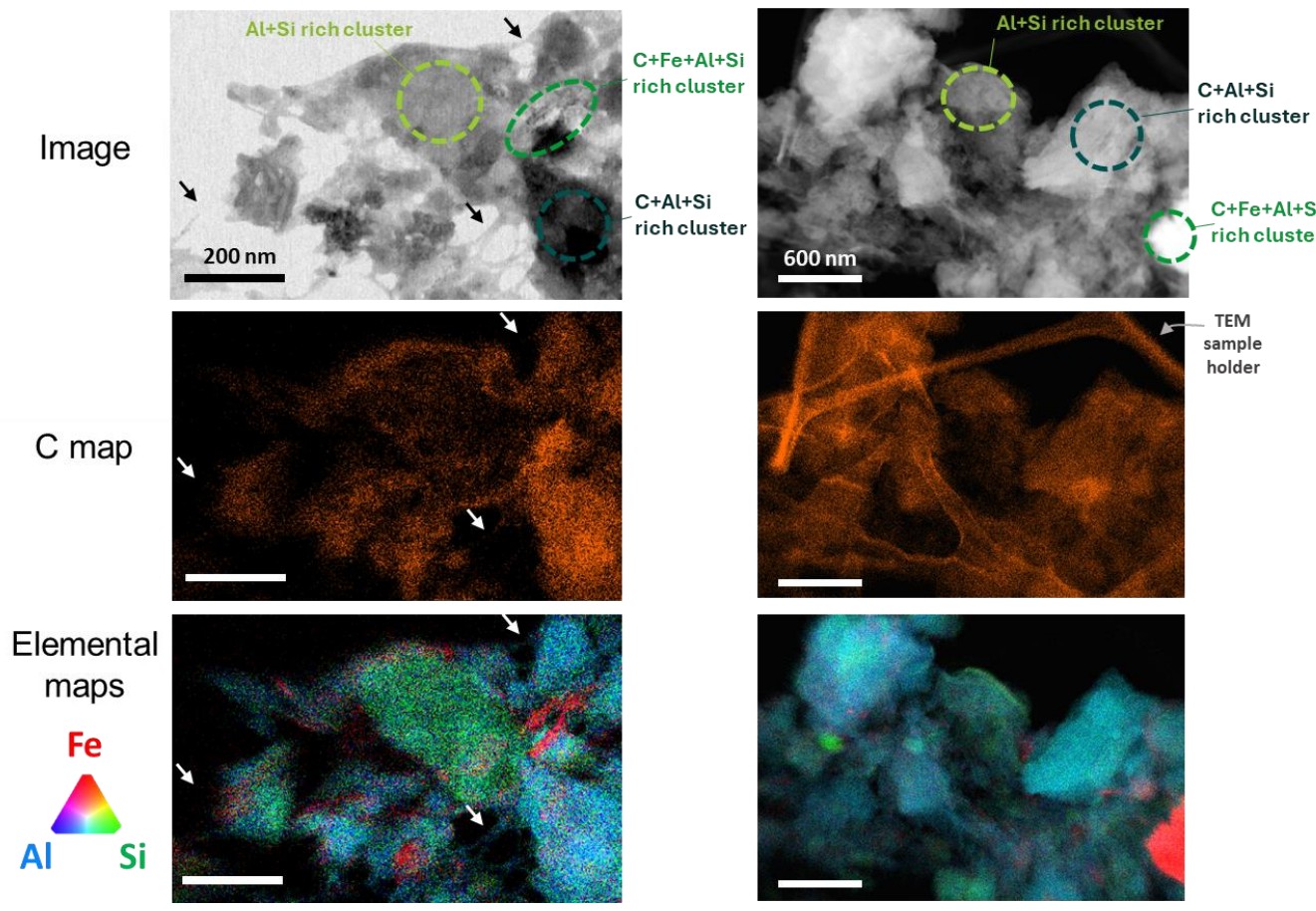

**Figure 5. Chemical mapping of forest and crop coprecipitates using STEM-EDX.** Chemical mapping of coprecipitates from the forest and the crop soil, showing imaged areas, carbon detections, and detections of Al, Si and Fe. Arrows indicate fibers characteristic of imogolites. See SI4 for additional mappings on forest and crop soils. The circled areas feature zones richer in C+Al+Si, C+Fe+Al+Si, and Al+Si (with less C).


### 3.2.3 Heterogeneity in the structure and composition of coprecipitates

After demonstrating the presence of a wide range of coprecipitate types in both soils, we further investigated the structural and chemical composition heterogeneity between the forest and crop coprecipitates. To do this we selected various regions on coprecipitates mappings and computed the atomic proportions within the selected area of ~200x200 nm (Fig. 6, see SI4 for area localization on maps). Atomic proportions of these regions revealed comparable average compositions of coprecipitates in both forest and crop Andosols (Fig. 6A). On average, coprecipitates comprised 35% C, 4% N, 5% Fe, 34% Al, and 22% Si. These results suggest that coprecipitates were mainly composed of organic molecules bound to an inorganic network of amorphous small oligomers up to proto-imogolites and less in the form of metal-organic complexes. Because according to atomic proportions, the organic/inorganic atomic ratio C/Al+Fe is 35/39, or 0.9. However, organic matter is in the form molecules containing a minimum of 8 to 10 C atoms, with 2 to 3 reactive sites per molecule able to bind to inorganic elements (by covalent or weak bonds). Then, if 3 reactive sites are present for 10 C atoms, the ratio of reactive sites to inorganic atoms (R/Al+Fe) is 0.3. Moreover, Si (not taken into account in this calculation) also competes with the reactive sites of Al and Fe (Lenhardt et al., 2023). According to the atomic proportions acquired, it is impossible for all organic matter to be bound to only a single Al or Fe monomer in the form metal-organic complexes. Hence, most organic molecules must be predominantly bound to oligomers (2-3 inorganic atoms made of Al, Fe and Si) in the form of nanoCLICs and proto-imogolites. Moreover, these averaged proportions masked underlying heterogeneities in atomic proportion within coprecipitates, including areas with high C+Al+Si proportions (averaging 44% C, 6% N, 2% Fe, 32% Al, and 17% Si), areas with high C+Fe+Al+Si proportions (44% C, 5% N, 14% Fe, 23% Al, and 13% Si), and areas with high Al+Si proportions but lower C proportions (21% C, 3% N, 2% Fe, 42% Al, and 32% Si; see Fig. 4E). Such heterogeneities were noted in both the forest and crop coprecipitates (Fig. 6B), indicating a consistent variation in elemental proportions within the coprecipitates. Following the same logic as above, C+Al+Si-rich and Al+Si-rich zones (with less C) can be attributed to zones containing mainly coprecipitates in the form of C+Al+Si nanoCLICs and proto-imogolite+OM. Concerning the C+Fe+Al+Si rich zones, additional coprecipitate forms may be present, such as C+Fe+Al+Si nanoCLICs and some Fe nanophases associated with organic matter. Therefore, all these forms of coprecipitate can be categorized as amorphous coprecipitates.

Subsequently, to determine whether the nature of organic matter could affect these heterogeneities (i.e., selective associations with elemental mix of Al, Si and Fe), we conducted elemental mapping for C, Al and Fe using scanning transmission X-ray microscopy (STXM; see Fig. 7A) and assessed the elemental speciation of carbon through C K-edge analyses (Fig. 7B). The elemental mappings for C, Al and Fe corroborated the findings from STEM-EDX and STEM-EELS analyses: specifically, a dominant co-localization of C with Al (as indicated by the purple color in Fig. 7A), with heterogeneities in regions of a few hundred nanometers (~500x500 nm), locally enriched in Fe (see areas 6, 7 and 8 in Fig. 7A), Al (area 5), or C (areas 3 and 4) observed in both the forest and crop soils. Overall, C speciation results exhibited multiple peaks indicative of aromatic-C (~285 eV), phenolic-C and ketonic-C (~286.6 eV), carboxylic-C (~288.4 eV), and carbonyl-C (~290.4 eV; Francis and Hitchcock, 1992; Cody et al., 1998; Boyce et al., 2002; Wan et al., 2007; Cosmidis and Benzerara,

2014; Le Guillou et al., 2018). Cluster analysis of the C K-edge did not reveal any distinct zones with different C speciation

(SI6), nor did it indicate any differences in C speciation between the forest and crop coprecipitates. Within the localized enriched area, a similar diversity of organic matter was detected in areas richer in C+Al+Fe (areas 1, 2, 8, 9), areas richer in C+Fe (areas 6, 7, 10), and areas richer in C+Al (area 5). Only the area enriched in C displayed distinct speciation, predominantly consisting of aromatic-C (~285 eV) and carboxylic-C (~288.4 eV). Moreover, compared to the area enriched in C, the speciation of C within coprecipitates richer in C+Al+Fe and C+Fe showed a higher pic at 286.6 eV attributed to

organic compounds made of phenolic-C and ketonic-C. These results indicated that a broad spectrum of organic molecules is present within amorphous coprecipitates of the forest and crop soils.

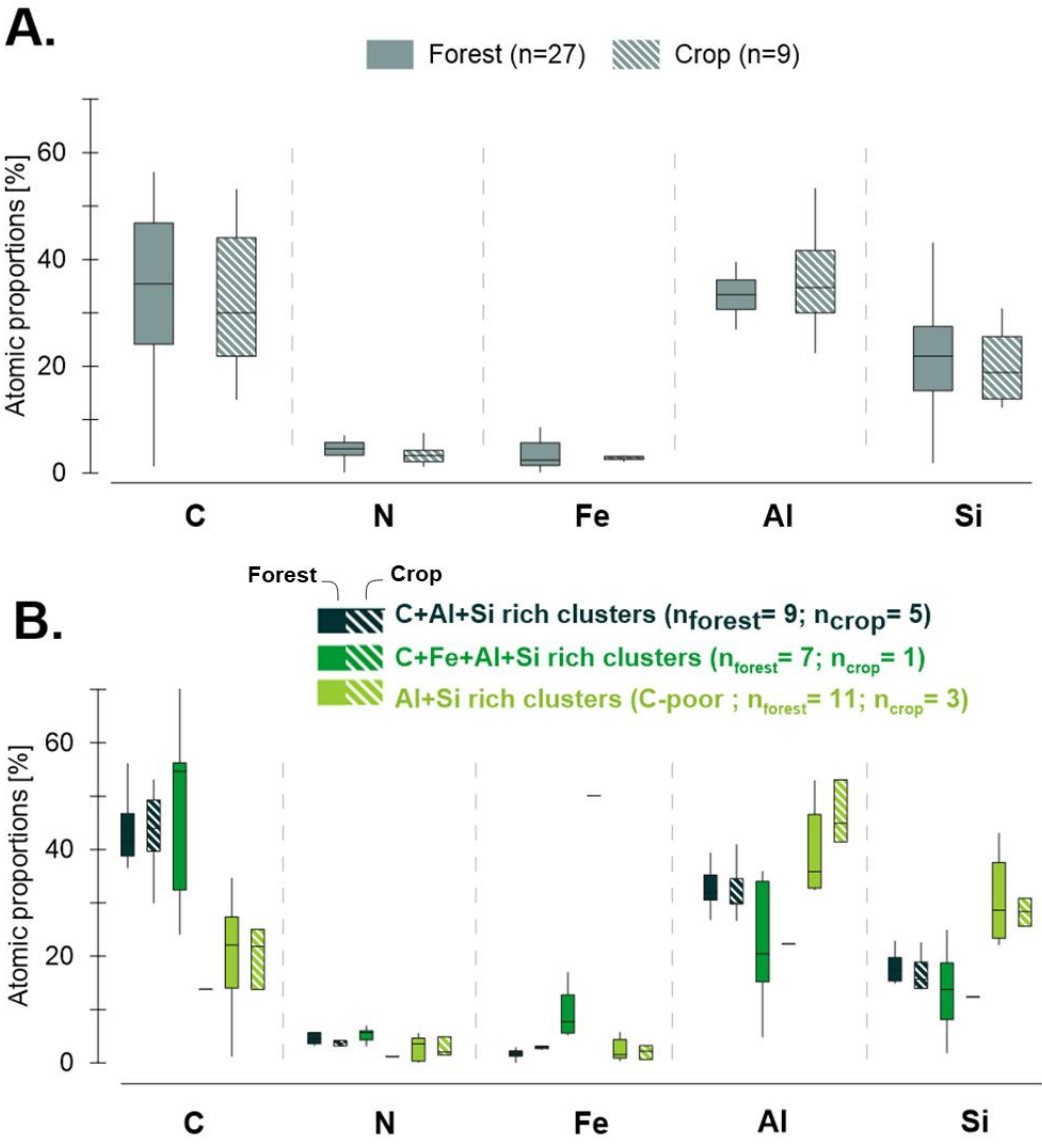

**Figure 6. Atomic composition of forest and crop coprecipitates based on STEM-EDX mapping.** From STEM-EDX mapping, various regions were selected to compute atomic proportions within the selected area of ~200x200 nm. (A) summarizes the average atomic proportions from selected areas across all mappings conducted on forest and crop soil coprecipitates, with 'n' representing the number of areas averaged. (B) details cluster-specific atomic proportions for clusters rich in C+Al+Si, C+Fe+Al+Si, as well as Al+Si (C poor), across various selected areas from all mappings on forest and crop soil coprecipitates. 'n' denotes the number of areas averaged. Examples of selected areas categorized as C+Al+Si, C+Fe+Al+Si, and Al+Si (C poor) are shown on Figure 5.

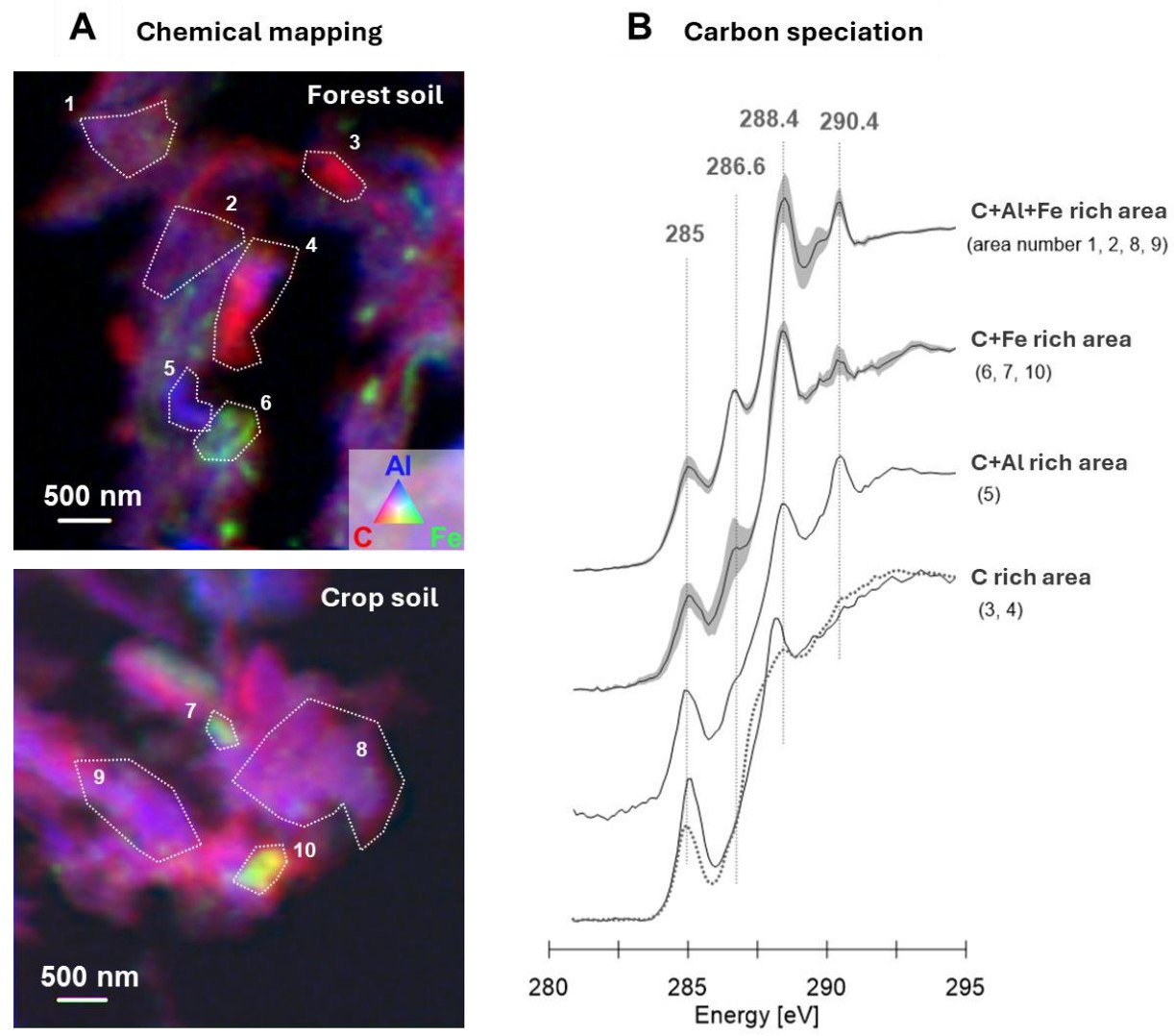

**Figure 7. Chemical mapping and organic matter characterization in coprecipitates using STXM mapping.** (A) STXM chemical mappings at the C K-edge, Fe L-edge, and Al K-edge of coprecipitates from forest and crop soils. (B) C K-edge spectra of the delineated area (outlined in panel A) showcase the principal energy bands associated with aromatic-C (~285 eV), phenolic-C and ketonic-C (~286.6 eV), carboxylic-C and C-OH (~288.4 eV), and carbonyl-C (~290.4 eV). C speciation exhibited consistency across the coprecipitates of both forest and crop soils (see individual spectra in SI6).

## 4 Discussion

### 4.1 Mineral-organic associations in Andosols:  a diversity of amorphous coprecipitates

Mineral-organic associations are typically conceptualized as organic matter adsorbed or coprecipitated with secondary minerals (Kleber et al., 2015, 2021). In Andosols, these associations are conceptualized as organic matter adsorbed on short-range ordered minerals such as imogolite, allophane, and ferrihydrite (Wada and Harward, 1974; Wada, 1985; Kleber et al., 2004; Parfitt, 2009). However, certain Andosols have been shown to lack these short-range ordered minerals (Levard et al., 2012), and nanoscale analyses revealed the presence of mineral-organic associations in the form of nanoCLICs (nanosized

coprecipitates of inorganic oligomers with organics), proto-imogolite+OM and some Fe nanophases+OM and metal-organic complexes (this study and in Jamoteau et al., 2023). More surprisingly, in this study, some short-range order minerals in the form of imogolite were observed, however, most of the C was in amorphous coprecipitates made of nanoCLICs, proto-imogolite+OM, some Fe nanophases+OM and some metal-organic complexes. These mix of coprecipitates are more amorphous and heterogeneous than previously proposed model of minerals in mineral-organic associations, with their

amorphous nature likely preserved by Si and organic matter that inhibit crystallization of short-range ordered minerals (Levard et al., 2012; Lenhardt et al., 2022, 2023). Mineral-organic association in form of amorphous coprecipitates are thus found in different Andosols: in an Andosol formed few 100 years ago on andesite parent material (Fe-poor parent material, this study), in an Andosol formed 40,000 years ago on basalt parent material (Fe-rich parent material, Jamoteau et al., 2023), and such amorphous mineral forms are likely present in many other Andosols, particularly young Andosols (e.g., Shimada et al., 2022).

These Andosols showed strong correlations between C and pyrophosphate-extractable metals (Alpp + Fepp), which primarily extracts metals from the least polymerized phases. However, before generalizing mineral-organic associations in the form of amorphous coprecipitates to all Andosols, future analyses must examine a broader range of Andosol types, considering various parent materials, ages and climates.

### 4.2 Heterogeneous chemical composition of amorphous coprecipitates

In these amorphous coprecipitates, our results showed an amorphous elemental mixture of C, Al, Si, and Fe. However, the proportions of these elemental mixtures showed enrichments in Al, Si, and Fe, with varying amounts of C, down to the hundred-nanometer scale (see Fig. 8A), likely forming C+Al+Si and C+Fe+Al+Si nanoCLICs, proto-imogolite+OM, some Fe nanophase+OM and metal-organic complexes. However, we propose that these phases should not be conceptualized as distinct entities. Instead, a continuum of phases should be considered, as illustrated by the ternary diagram in Fig. 8B. This diagram

shows distinct entities comprising (1) nanoCLICs enriched in C+Fe+Al+Si, (2) nanoCLICs enriched in C+Al+Si and proto-imogolites+OM, and (3) nanoCLICs enriched in C+Si+Al (Si rich) along with their intermediate forms. In addition to these predominant forms, the small amount of Fe in coprecipitates could also be present in form of (4) Fe nanophases+OM and (5) metal-organic complexes. This compositional heterogeneity in the mineral (or inorganic) portion of the coprecipitates may arise from their formation in microsites with locally diverse elemental solutions. For instance, this could be influenced by the

proximity to certain minerals that release specific elements into the soil solution, a process potentially controlled by microbial activity (Uroz et al., 2009; Bonneville et al., 2011, 2016). An alternative hypothesis for the compositional heterogeneity relates to the nature of organic matter, which might bind preferentially to certain elemental mixtures of Al, Si, and Fe. Our findings demonstrate that in overall, the organic matter speciation was diverse (composed of aromatic-C, phenolic-C, ketonic-C, carboxylic-C, and carbonyl-C) and consistent across areas down to the hundred-nanometer scale (Fig. 7). This finding aligns

well with the diversity organic matter observed in mineral-organic associations from various temperate and tropical soils, despite distinct vegetation compositions and soil mineralogy (Kinyangi et al., 2006; Lehmann et al., 2008; Solomon et al., 2012; Asano et al., 2018), and demonstrate that broad spectrum of organic matter can form mineral-organic associations in the form of amorphous coprecipitates. However, in areas richer in C+Al+Fe and C+Fe, a higher proportion of organic compounds made of phenolic-C and ketonic-C were observed (Fig. 7), suggesting a correlation between the presence of Fe in coprecipitates

and phenolic and/or ketonic-rich compounds. This correlation could be explained by a preferential binding of phenolic compounds with Fe, highlighted in the literature (Schmidt et al., 2013; Mimmo et al., 2014). Moreover, the enrichment in aromatic-C in C rich area (not associated with other elements) aligns with findings in other soil types (Solomon et al., 2012; Lutfalla et al., 2019), and may stem from the presence of particulate organic matter, thus indicating more degraded organic matter in coprecipitates. In conclusion, these results highlight the complex and heterogeneous nature of amorphous

coprecipitates in Andosols. These findings underscore the importance of considering a continuum of coprecipitate phases rather than distinct entities to better understand the interactions between organic matter and amorphous mineral components in soils.

## A. Heterogeneous composition of mineral-organic associations

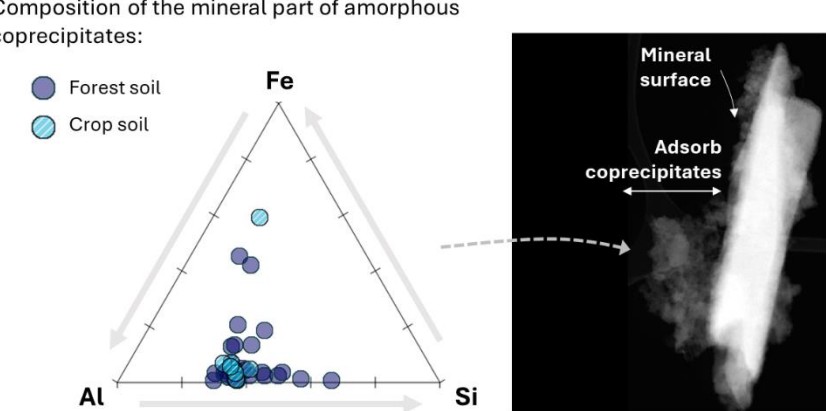

## B. Conceptual model of mineral-organic associations in Andosols

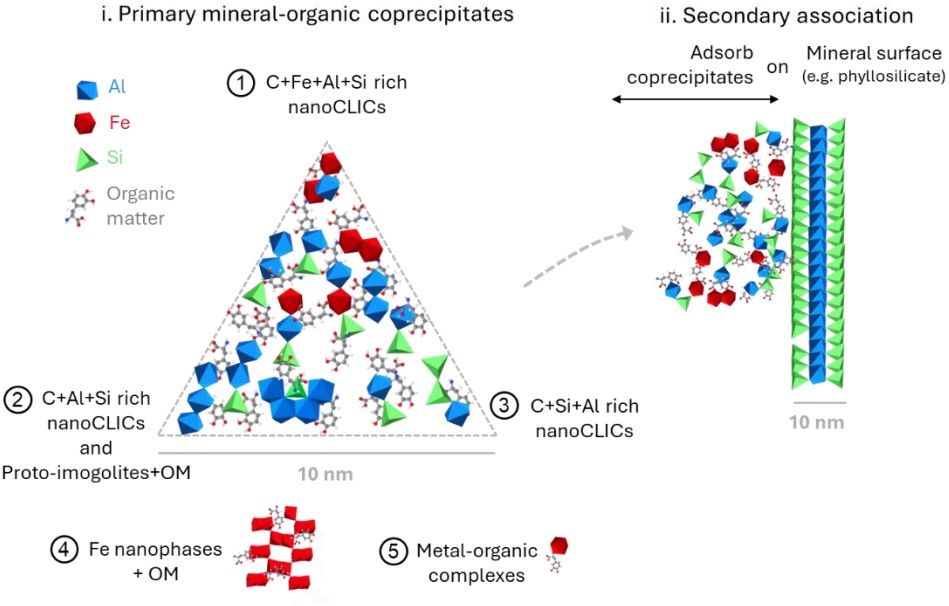

**Figure 8. Structure and composition of mineral-organic associations in Andosols.** (A) Atomic proportions of Al, Si and Fe in mineral-organic associations in the amorphous phase derived from STEM-EDX mappings. The right panel image illustrates the type of area analyzed by STEM-EDX, highlighting regions where amorphous coprecipitates have been found adsorb on crystal's surfaces. (B) Conceptual model of primary mineral-organic associations in Andosols. i. Primary mineral-organic coprecipitates comprising a mix of nanoCLICs, proto-imogolites+OM (numbers 1, 2 and 3), Fe nanophases+OM (4) and metal-organic complexes (5). The ternary diagram depicts the compositional heterogeneity of these amorphous coprecipitates and the continuum of forms between C+Fe+Al+Si rich nanoCLICs (1), C+Al+Si rich nanoCLICs and proto-imogolites (2) and C+Si+Al rich nanoCLICs (3). Additionally, a conceptual model of secondary association is presented, showcasing the interaction of primary mineral-organic coprecipitates with mineral surfaces (e.g. 2:1 phyllosilicate).

## 4.3 A secondary interaction: adsorption of amorphous coprecipitates onto mineral surfaces

In prevailing conceptual frameworks of mineral-organic associations, two dominant paradigms are recognized: coprecipitation and adsorption of organic matter with minerals (Kleber et al., 2015, 2021). Our investigations reveal that amorphous coprecipitates represent the main form of mineral-organic associations in some Andosols. However, a secondary association was observed : adsorption of amorphous coprecipitates onto mineral surfaces (Fig. 3 and 8), or within phyllosilicate layers (refer to Fig. SI2) and likely adsorb onto short-range order surfaces (e.i. imogolite, Fig. 5). These findings suggest that coprecipitation processes likely occur either upstream or simultaneously with adsorption processes on mineral surfaces. These processes can take place in soil solutions, which contain both organic compounds and elements resulting from mineral weathering (such as Si, Al, Fe, Ca, and Mn; Campbell et al., 1989; Giesler and Lundström, 1993; Manderscheid and Matzner, 1995; Strobel et al., 2001; Kaiser et al., 2002). Coprecipitation and subsequent adsorption on mineral surfaces are, therefore, interrelated phenomena. This study highlights that mineral-organic associations need to be conceptualized as a sum of subsequent interactions rather than a single interaction (as amorphous coprecipitates being adsorbed onto mineral surfaces) thereby illustrating the complexity of mineral-organic associations assemblages in soils. Moreover, this secondary association between coprecipitates and mineral surfaces aligns with studies demonstrating the binding capacity of mineral-organic associations in Andosols (Wagai et al., 2018; Asano et al., 2018; Shimada et al., 2022). This binding capacity for associations between mineral-organic compounds and fine soil minerals could contribute to the structuration of aggregates, particularly clay-size aggregates (Asano et al., 2018; Wagai et al., 2018; Shimada et al., 2022). In sum, our results underscore the complexity of primary and secondary mineral-organic associations in soils, highlighting the interplay between coprecipitation and adsorption processes. These interactions are crucial for soil organic matter stabilisation and the formation and stability of soil aggregates.

## 4.4 Differences between the forest and crop soils: toward a quantitative loss of amorphous coprecipitates rather than changes in mineral-organic association types

The transition from natural to agricultural systems typically results in a C stock decline (Poeplau and Don, 2013; Sanderman et al., 2017). In the studied Andosols, the crop soil had 46 % less C stock compared to the forest soil on 0-30 cm depth (Fig. 2), consistent with previous findings following forest-to-crop conversions in the literature (Poeplau and Don, 2013). This substantial C difference showed up to 75% less C in mineral-organic association in the crop topsoil compared to the forest topsoil. However, the same types of mineral-organic associations were present in both Andosols: nanoscale examination exhibit amorphous coprecipitates characterized by similar inorganic structural heterogeneity and organic matter diversity (Fig. 5-7). Moreover, pyrophosphate extractions on bulk soils highlighted a twofold difference in extracted amount of Al, Si, and Fe between the forest on the crop Andosols, suggestive of less amorphous mineral phases in the crop soil (Rennert, 2018; Rennert and Lenhardt, 2024). These extractions should specifically target the amorphous coprecipitates and imogolites observed at the nanoscale (Fig. 4) in this context. Compared to the forest topsoil, the lower amount of C within mineral-organic associations simultaneous to the lower quantity of amorphous and poorly crystalline minerals in the crop topsoil suggests a

lower abundance of amorphous coprecipitates in the crop topsoil. Furthermore, although the total C content in the crop topsoil was lower, the relative C proportion within the amorphous coprecipitates remained similar to the ones in the forest topsoil (Fig. 6, within the limits of the analyzed areas), further indicating less amount of amorphous coprecipitates in the crop topsoil (and not only a lower C content within the amorphous coprecipitates). Ultimately, these results suggest that the cropland conditions mainly affected the amount, rather than the type, of mineral-organic associations in the studied Andosol. However, it is important to consider that the representativeness of a forest-to-cropland conversion in this study has some limitations: firstly, the samples come from only one soil profile taken at a single point in time, which means they may not fully capture the spatial variability within a plot. Secondly, the quantitative differences in organic carbon content and extractable pools are based on a single observation per depth. Although multiple depths within a profile contribute valuable information to the overall pattern, they do not function as replicates per se.

### 4.5 Insights into the stability of mineral-organic associations in Andosols

In the literature, mineral-organic associations are considered to be stable over centennial durations (Trumbore et al., 1989; Bol et al., 2009; Feng et al., 2016; Shimada et al., 2022). However, in Andosols, which are rich in mineral-organic associations, just a few decades of cultivation can lead to the destabilization of C (Verde et al., 2005; Basile-Doelsch et al., 2009; Osher et al., 2003; Dube et al., 2009; Koga et al., 2020). In this study, if the two analysed Andosol profiles represent decreases between forest and crop soil (see the limits of sample representativeness of forest-to-cropland conversion in 4.4), our results suggest that (i) the observed difference in C content can be primarily due to the disruption of mineral-organic associations within the 0-30 cm depth of the Andosol, and (ii) mineral-organic associations in the form of amorphous coprecipitates may be prone to destabilization after three decades of agricultural activity. This indicates that while such associations may persist for a long time within certain Andosols, they can be susceptible to disruption contingent upon agricultural practices. This potential disruption could be linked to the disruption of amorphous mineral constituents of coprecipitates, namely Al, Si, and Fe—given that amorphous Al and Fe exhibit diminished stability under reduced pH conditions. Consequently, amorphous coprecipitates may experience partial solubilization as a result of the pH diminution in the crop soil, potentially leading to the loss of these associations. Despite a total pH difference of approximately one unit, from 6.3 in the forest soil to 5.6 in the crop soil (Fig. 2), this overall lower pH in the crop soil could reflect even lower pH driven by root exudates and the biological activity (Keiluweit et al., 2015; Bernard et al., 2022), potentially leading to coprecipitates disruption. Moreover, given the heterogeneous composition of amorphous coprecipitates (Fig. 8), their transformations could have varied vary based on local composition. For example, amorphous coprecipitates with a high Fe content may exhibit increased sensitivity to redox fluctuations and pH decreases, while those enriched with Al and Si may show sensitivity to pH changes. Such compositional-dependent transformations were not found in our results, which demonstrated consistent heterogeneity of amorphous coprecipitates across both forest and crop Andosols (Fig. 4). In addition to their chemical vulnerability to physicochemical changes, agricultural activity may also play a role in amorphous coprecipitates disruption by lessens aggregate occlusion, enhancing interaction between the microbial community, roots, and coprecipitates (Bailey et al.,

2019). Consequently, if the two analysed Andosol profiles represent decreases of mineral-organic associations between the forest and crop soils, this study would suggest that mineral-organic associations in the form of amorphous coprecipitates could be prone to disruption due to agricultural conversion.

## Conclusion

455     The investigation of mineral-organic association types down to the micro and nanoscale in the studied Andosol demonstrated the presence of amorphous coprecipitates, made of a mix of nanoCLICs, proto-imogolite+OM and some Fe nanophases+OM and metal-organic complexes. These amorphous coprecipitates observed in both the forest and crop soils exhibited the same amorphous composition and chemical heterogeneity, challenging prior conceptualization of mineral-organic associations in Andosols by demonstrating the presence of amorphous coprecipitates rather than solely organic matter
460     associated with short-range order minerals, such as imogolite and allophanes. The organic matter composition of amorphous coprecipitates was diverse, made of aromatic-C, phenolic-C, ketonic-C, carboxylic-C and carbonyl-C, suggesting the potential for coprecipitates formation with various types of organic matter. Moreover, our spatial mapping suggests that these amorphous coprecipitates can adhere to mineral surfaces (e.i., onto phyllosilicates and imogolites), suggesting that such associations are a composite of multiple interactions rather than a singular form. These results underscore the complexity of primary mineral-
465     organic associations and their subsequent interactions in soils, highlighting the interplay between coprecipitation and adsorption processes. These interactions are crucial for soil organic matter stabilisation and the formation and stability of soil aggregates, particularly at the clay-size level. While the crop topsoil was observed to have 75% less C in mineral-organic associations than the forest topsoil, alongside notable physicochemical differences, including a lower pH, the presence of similar amorphous coprecipitates in both forest and crop soils was confirmed. Although the sample size for comparing land-
470     use types is limited, these differences did not seem to alter the nature of mineral-organic associations or the C content within the amorphous coprecipitates but rather suggest to affect the amount of amorphous coprecipitates in the crop topsoil. This study demonstrates the crucial role of amorphous coprecipitates in C stabilization in Andosols but also suggest their vulnerability to disruption after 30 years of agricultural conversion, thereby challenging our understanding of the persistence of mineral-organic associations in Andosols.

475 **Data availability**

Data are available upon request from the corresponding author.

## Author contributions

F.J. contributed to conceptualization, methodology, formal analysis, and writing – original draft. E.D. participated in conceptualization, methodology, formal analysis, writing – review and editing. N.C. and C.L. were involved in conceptualization, methodology, and formal analysis, writing – review and editing. T.W., F.S.A., S.S, A.D., D.B., M.L.P., P.C., V.V., and N.B. contributed to sampling, methodology, and formal analysis. I.D.B. participated in conceptualization, methodology, formal analysis, and writing - review and editing.

## Competing interests

The authors declare that they have no conflict of interest.

## Acknowledgments

The author thanks the research funding partners: ANR (NanoSoilC project ANR-16-CE01-0012-02), the Equipex nanoID platform (2010-2019), la Région SUD and CIRAD (Emploi Jeunes Doctorants, subvention n°2019_03559, DEB 19-574). The author thanks Stefan Stanescu for his support during analyses on the HERMES beamline at the Soleil synchrotron and Marco Keiluweit for his edits to the manuscript. The author thanks PiCSL-FBI core platform (IBDM, AMU-Marseille, member of the France-BioImaging national research infrastructure, ANR-10-INBS-04), where the cryo-sections were conducted.

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
