# Peer review of "Interplay of coprecipitation and adsorption processes: deciphering amorphous mineral-organic associations under both forest and cropland conditions"

_EGUsphere, 2024_

## Author Comment (AC1)

Supplementary figures for review :

[Figure]

*Figure 1. Picture of the forest soil profile during sampling.*

**Area 1**

[Figure]

**Area 5**

[Figure]

*Figure 2. Deconvolution procedures applied to the C K-edge spectra of area 1 rich in C+Fe+Al and area 5 rich in C+Al. In this figure, the energy shift is not corrected, which explains why the peak at 286.6 in figure 7 is slightly shifted in energy in this figure.*

---

## Author Response (AR1)

**Author response to Editor's comments:**

**Public justification (visible to the public if the article is accepted and published):**

I appreciate the authors' thorough responses to the reviewer comments, and support the proposed revisions. However, there remains a significant issue with the stated aim and interpretation of the results given the limitations of the experimental design that is not adequately addressed in the author responses.

The reviewers both expressed concern regarding the interpreted effects of land use change, given that only single soil pits were used for this study. The authors responded in one such instance that, "Data now encompass not only the 10-20 cm horizon but the entire soil profiles (0-90 cm; 18 samples now), which greatly enhances the accuracy of the forest-to-crop conversion impact assessment", but this is not the case. These remain single pits from each site (with samples taken from three sides but composited prior to analysis), so it is not possible to estimate spatial variation in soil properties within each site and therefore difficult to make a reliable contrast between the two land uses. Further, as pointed out by Reviewer 2, there are no repeat samples through time and the validity of the space-for-time approach using these sites to infer causation is not clearly established. Even if it were, the deep tillage carried out in the cultivated soil with its associated mixing and redistribution of soil material calls into question the direct comparability of horizons across the land uses, making it even more difficult to directly assess C losses and associated mechanisms (it could be redistribution rather than loss). Given the limitations of the experimental design, any claims of causal effects of land use conversion (and "change") are highly speculative and should not be overstated.

In a revised manuscript, definitive claims about effects of land use conversion such as, "the agricultural conversion has led to a significant decrease of 75% C mainly in form of amorphous coprecipitates (on 0-30 cm depth)" should be removed or softened significantly to reflect the inability to assess causal effects given the experimental design. Statements regarding change over time (e.g., "decrease", "loss", "change") should also be reworded to reflect the limitations in the space-for-time approach. For example, the previously quoted sentence could be reworded to, "the cropland soil profile was observed to have 75% less C than the forest soil profile, with the largest differences in the form of amorphous coprecipitates (in the 0-30 cm depth)." The aim of the paper to do with contrasting land use types (forest vs. cropland) should be softened (e.g., rather than "determine", something like "investigate", "explore", "probe", etc.) and the major caveat of having no estimate of variation within each land use type must be presented up front and reiterated throughout the manuscript wherever differences are presented. Differences between the two profiles may be discussed, but differences between the land use types should only be suggested (they warrant further research with an appropriate experimental design).

Both reviewers suggested focusing the manuscript less on the effects of land use conversion and more on the compositional and structural insights regarding the nature of these organo-mineral associations, which I agree with.

Author's answer: We appreciate your acknowledgment of our thorough responses to the reviewer's comments and your support for the proposed revisions. We have made the requested changes. We have revised our interpretation of the results to avoid overstating any causal effects of land use conversion. We hope these modifications meet your expectations.

More specifically, these changes include:

i.   The aims have been clarified as follows: "*This study aims to (i) investigate the predominant mineral-organic associations within an andosol (developed on Fe-poor parent material) under both forest and cropland conditions, and to explore possible differences in the types of mineral-organic associations, and (ii) to determine if mineral-organic associations in a forested Andosol developed on Fe-poor parent material (andesite) are similar to mineral-organic associations in a forested Andosol developed on Fe-rich parent material (basalt; from Jamoteau et al., 2023).*"

ii.  We acknowledge that using single soil pits for each site limits our ability to estimate spatial variation in soil properties and to make a reliable comparison between the two land uses. The result now highlights "difference between the forest and crop soil profiles/coprecipitates" rather than "forest-to-crop conversion impacts". This limitation is now recognized in interpretation paragraphs 4.4 and 4.5. See, for example the concluding sentences of these paragraphs as follows : "*Ultimately, even though the data are from only two soil profiles, if they represent a forest-to-crop conversion, our results suggest this conversion mainly affects the amount, rather than the type, of mineral-organic associations in the studied Andosol.*" and "*Consequently, if the two analyzed Andosol profiles represent a forest-to-crop conversion, this study suggests that mineral-organic associations in the form of amorphous coprecipitates can be prone to disruption due to agricultural conversion.*".

iii. Additionally, we have also considered your point and Reviewer 2's point about the impact of tillage on soil material distribution. In order to ensure a quantitative difference in C stock between the two topsoil profiles, despite the possible redistribution of C by ploughing, we have included the cumulative carbon stock data in equivalent mass in Figure 2. The data indicates that the accumulated stock over the 0-40 cm depth (maximum ploughing depth) is half as much in cultivated soil compared to forest soil, demonstrating a difference in C stock between the forest and crop topsoils.

iv.  Furthermore, regarding the manuscript title, we have decided to modify the original title to: 'Forest-to-crop conversion reduces quantities of mineral-organic associations in the form of amorphous coprecipitates.' Moreover, upon request, we are willing to modify it to include more conservative language and incorporate 'may' as follows: "Forest-to-crop conversion may reduce quantities of mineral-organic associations in the form of amorphous coprecipitates."

**General Comments of Anonymous referee #2**

**Overview Comments:**

Jamoteau et al. present a variety of high spatial-resolution analyses probing organic and mineral phases in a Fe-poor Andosol, focusing specifically on distribution of C within either SRO or amorphous phases. The imaging and spectroscopy approaches are strong and presented well (though with some methodological detail expansion needed), and in particular, the finding regarding secondary interactions between an SRO phase and associated (potentially adsorbed) nanoCLIC phases is quite an interesting result! The contrast between a forested and cultivated setting is based on single-point measurements (based on my understanding of the methods and data presented), which means that some of the overall interpretations regarding land-use impacts need to be more constrained in their presentation. In addition, some broader context for prior work focusing on amorphous phases in Andosols would help strengthen the knowledge gap addressed. Examples of these considerations are outlined in the detailed comments below.

Author's answer to general comments: The authors thank the reviewer for their valuable insights and comments, which have greatly improved the manuscript. We are grateful for the time and effort the reviewer has dedicated to this work.

According to the comments received, major changes have been made to the manuscript:

(1) Particularly in the nomenclature of mineral-organic associations. The nomenclature is now clarified in the introduction section, and a related figure illustrates the forms of associations (Fig. 1), which greatly enhance the understanding of the manuscript. Additionally, following the reviewer's comments, the interpretation of mineral-organic association forms has been revised to include a diversity of associations composed of a mix nanoCLICs, proto-imogolites+OM, and some Fe nanophases + OM and metal-organic complexes (see in the text sections 3.2 to 4.2 and Figure 8)

(2) Additionally, significant changes have been made regarding the comparative differences in C concentration, pH, and mineralogy between the forest and cultivated soils. Data now encompass not only the 10-20 cm horizon but the entire soil profiles (0-90 cm; 18 samples now), which greatly enhances the comparison between both soil profiles. We acknowledge that using single soil pits for each site limits our ability to estimate spatial variation in soil properties and to make a reliable comparison between the two land uses. This limitation is now recognized in interpretation paragraphs 4.4 and 4.5. See, for example the concluding sentences of these paragraphs as follows : "Ultimately, even though the data are from only two soil profiles, if they represent a forest-to-crop conversion, our results suggest this conversion mainly affects the amount, rather than the type, of mineral-organic associations in the studied Andosol." and "Consequently, if the two analyzed Andosol profiles indeed represent a forest-to-crop conversion, this study suggests that mineral-organic associations in the form of amorphous coprecipitates may be prone to disruption due to agricultural conversion.".

(3) Additionally, the results of the secondary interaction between amorphous coprecipitates and mineral phases have been further emphasized. This result is now mentioned in the abstract, the discussion has been expanded into section 4.3, and Figure 8 now illustrates this secondary association.

Author's answer to detailed comments:

-Line 20: It's unclear what is being referred to in terms of "predominant mineral-organic associations." In a specific system, or croplands in general?

Author's answer: We agree that the original wording was unclear. We have revised the sentence as follows: "Here, we aimed to investigate the prevailing forms of mineral-organic associations and explore possible differences in these association types between a forest and a crop soil.".

-Line 22: It is important to establish upfront how the contrast between the forested and cultivated area is being made: are these two adjacent areas on the same site? In other words, is the forested area a fair reference point?

Author's answer: We have clarified that the forested and cultivated areas are 300 meters apart as follows: "To achieve this, we collected andosol samples from both a forested area and a cultivated area, located 300 meters apart."

-Line 22: As with any study contrasting management or ecosystems, it is important to justify that differences are due to cultivation per se (rather than other potential correlated factors). Is the contrast specific to cultivation (i.e., tillage) or overall conversion from forest to agricultural systems, which alters input type, root systems, organic surface layers, etc.

Author's answer: We agree that the wording was unclear, and our study focuses on a forest-to-crop conversion rather "cultivation types". We changed the wording "cultivation" to "conversion from forest to agricultural systems" or "forest-to-crop conversion" or "agricultural conversion" throughout the manuscript. Additionally, in the discussion sections we acknowledged that using single soil pits for each site limits our ability to estimate spatial variation in soil properties and to make a reliable comparison between the two land uses. This limitation is now recognized in concluding sentences of paragraphs 4.4 and 4.5.

-Line 26: "Down to a few... " What does this refer to specifically? A few hundred nanometers in spatial resolution? NanoCLIC "features" a few hundred nanometers in size? Some clarification needed here.

Author's answer to comment L26 and 27: We have significantly revised the abstract, and the results now appear as follows: "At the micro and nanoscale spatial resolution, we observed mineral-organic associations in the form of amorphous coprecipitates, composed of a mix of C+Al+Si and C+Al+Fe+Si nanoCLICs, proto-imogolites and organic matter and some Fe nanophases associated with organic matter and metal-organic complexes. This challenges prior conceptions of mineral-organic associations in Andosols by demonstrating the presence of amorphous coprecipitates rather than solely organic matter associated with short-range order minerals (e.i. imogolite and allophanes)."

-Line 27: The wording describing these distinct elemental compositions is unclear and I think could be presented more simply. My interpretation from is that there are three primary regions with compositions of (1) C+Fe+Al+Si, (2) C+Al+Si, or (3) Al+Si. Is that correct?

Please refer to our answer to comment L26.

-Line 28: "Exhibited various C..." Not sure I follow what is meant by this statement. Does this imply that organic matter speciation (rather than quantity) is not tied to the distribution of other elements, but occurs randomly throughout the soil samples/NanoCLICs?

Author's answer: Following the major changes made to the abstract, we have removed this sentence.

-Lines 28-30: Since the phrasing of the previous two sentences describing results is unclear, I'm not sure I follow exactly what they indicate amorphous coprecipitate-like structure rather than surface associations with crystalline minerals. There seems to be an indication that these elemental compositions/distributions and C speciation are in line with coprecipitates specifically, but the direct connection between the two is unclear.

Author's answer: We thank the reviewer for their comment. This section has been modified as follows: "Moreover, chemical mappings suggested that these amorphous coprecipitates may adhere to mineral surfaces (i.e. phyllosilicates and imogolites), revealing subsequent interactions of mineral-organic associations in soils."

-Lines 30-31: How were NanoCLICs quantified? A bit more information is needed in the abstract to be able to assess this conclusion (and based on the single measurements, as I interpret it, this value needs to be considered with caution).

Author's answer: Following the major changes made to the manuscript, these results are now communicated as follows: "While the presence of similar amorphous coprecipitates in both the forest and crop Andosols was confirmed, the crop soil had 75 % less C in mineral-organic associations (in the 0-30 cm depth), suggesting that despite quantitative differences in mineral-organic associations, their nature was identical.".

-Line 31: I am not sure that the conceptualization as a "nanoCLIC" is fundamentally different than the description of amorphous organo-mineral materials in Andosols, which has significant historical focus. The observations and description of nanoCLIC composition and structure is novel information, but there is historical precedence for considering Andosol organo-mineral associations as amorphous phases (perhaps within a range of different related processes, such as organo-metal complex formation and precipitation, e.g. as discussed in Wada 1985 (https://link.springer.com/chapter/10.1007/978-1-4612-5088-3_4). I suggest highlighting the compositional and structural insights from the study more clearly in the abstract (such as interactions between amorphous phases and SRO phases, which I thought was really interesting!) to highlight more clearly how this improves our understanding of their persistence and vulnerability.

Author's answer: We thank the reviewer for their comment.  Following the major changes made to the manuscript, the interpretation of mineral-organic association forms has been revised to include a diversity of associations composed of a mix nanoCLICs, proto-imogolites+OM, and some Fe nanophase + OM (see in the text sections 3.2 to 4.2 and in Figure 8). For the cases of organo-metallic complexes, these forms are now mentioned in the introduction (see Fig.1) and mentioned in the results and discussion part (see section 3.2.3 and lines 275-290). According to our results, the coprecipitates are mainly composed of C associated more with oligomers rather than with simple monomers (see lines 275-290). However, organo-metallic complexes are not completely excluded and are represented in the final Figure 8, but they do not seem to be the majority of coprecipitate forms present in this andosol.

Regarding the secondary interaction between amorphous coprecipitates and minerals (phyllosilicates or short-range ordered minerals), this result is now mentioned in the abstract, as follows: "Moreover, chemical mappings suggested that these amorphous coprecipitates may

adhere to mineral surfaces (i.e. phyllosilicates and imogolites), revealing subsequent interactions of mineral-organic associations in soils."

-Line 32: Please clarify what is meant by crop-induced. Is the effect one from cultivation (tillage), a particular crop, being an agricultural system in general...? It is not clear what system is being contrasted.

Author's answer: Following Reviewer#2 and Editor's comment, we rephrased the sentence as follows: "While the presence of similar amorphous coprecipitates in both the forest and crop Andosols was confirmed, the crop soil had 75 % less C in mineral-organic associations (in the 0-30 cm depth), suggesting that despite quantitative differences in mineral-organic associations, their nature was identical.".

-Line 45: Coprecipitation of "amorphous metal-organic complexes" is also a well-known variant of coprecipiation (which doesn't necessarily fall within the "short-range ordered" mineral phases), e.g. as mentioned in Chen et al. 2014 (https://doi.org/10.1021/es503669u) and references within, and mentioned in Kleber 2015 chapter cited. While most of the mentioned studies focus on laboratory coprecipitation experiments, I think these concepts align well with observations (some spatially resolved, some not) of "amorphous" organo-mineral associations in a variety of "coprecipitation" environments - e.g., spodic horizons: there is extensive literature related to podzolizaiton which is inherently a coprecipiation process, e.g. see Buurman and Jongmans 2005, https://www.sciencedirect.com/science/article/abs/pii/S0016706104001855). In all, I think that the premise that coprecipitation is a traditionally a dominantly SRO mineral-related process is not well-supported, and hence the distinction of a "nanoCLIC" from "amorphous organo-metal complex" is less clear. I don't think the two are synonymous - a "complex" implies a particular binding mechanism and molecular arrangement that I think is more ambiguous with the term nanoCLIC - but I think it is important to put this concept in the context of prior conceptualizations of amorphous organo-mineral coprecipitates.

Author's answer: We agree that the wording was unclear. We have clarified the beginning of this paragraph and added a figure on the types of mineral-organic associations and their characteristics (see Figure 1). We have also rephrased this paragraph to better integrate it into the existing literature on the subject. The paragraph now better synthesizes the prevalent types of associations in soils with acidic to neutral pH, including SRO+OM, proto-imogolites+OM, nanoCLICs, and metal-organic complexes.

Regarding the difference between nanoCLICs and metal-organic complexes, we have added this sentence to the paragraph: "These nanoCLICs phases do not fit into the spectra of metal-organic complexes because they are more heterogeneous in composition (Al, Fe, Si, and some Ca, Mg, K, etc.) and organic molecules are linked to metallic oligomers of approximately 2-3 atoms (Tamrat et al., 2019; Jamoteau et al., 2023)." Furthermore, we do think that the binding in nanoCLICs is unclear; the organic matter can be weakly or covalently bound to Al, Fe, and to a lesser extend Si, or trapped in the amorphous skeleton made of Al, Fe, and Si, whereas in metal-organic complexes, only metal-organic interactions are mentioned. We believe that the additions to the paragraph and Figure 1 will help the reader better picture this difference.

-Lines 50-51: These studies cited prior (Kinyangi, Wan, Solomon, etc.) do not to my knowledge speak to stability under agricultural practices - it should be made more clear that this is a potential application that is being leveraged in this study.

Author's answer: We rephrased the sentence as follows: "Recent advances in nanoscale (spectro)microscopy (e.g., TEM-EDX, STXM) have facilitated precise analyses of mineral-organic associations' composition and structure (Kinyangi et al., 2006; Solomon et al., 2007; Wan et al., 2007; Solomon et al., 2012; Asano et al., 2018), offering deeper insights into mineral-organic associations composition."

-Lines 52-53: The link between C stabilization by SRO minerals to the data presented by the cited study is not clear.

Author's answer: This part of introduction has been modified as follows: "In Andosols, i.e. soils with high concentrations of mineral-organic associations, microscopy and spectroscopy analyzes have refuted the stabilizing role of short-range order minerals in the form of imogolite or allophane for C (Levard et al., 2012). Instead, organic C was primarily associated in the form of nanosized coprecipitates of inorganic oligomers with organics (nanoCLICs). In such structures, organic molecules are linked to a few atoms of Al, Fe, or Si without crystalline structures (Tamrat et al., 2018, 2019; Jamoteau et al., 2023). [...] These findings challenge previous assumptions about the types of mineral-organic associations in andosols developed from basalt parent material. Suggesting that, in some situations, Andosol's mineral-organic associations may contain a more amorphous structure than earlier proposed models of short-range ordered minerals (Jamoteau et al., 2023).

-Lines 55-57: This statement seems like a good summary of the key question. However, there are a number of potential considerations here in terms of variation among Andosols (age and development, for example, e.g. in Torn et al. 1997 and many other variations) - I think some qualifying language here is needed (e.g., "in some situations Andosols may have a more amorphous constitution than earlier proposed models" or similar).

Author's answer: We do agree that age development is a factor to consider, we modified the statement as follows: "These findings challenge previous assumptions about the types of mineral-organic associations in andosols developed from basalt parent material. Suggesting that, in some situations, Andosol's mineral-organic associations may contain a more amorphous structure than earlier proposed models of short-range ordered minerals (Jamoteau et al., 2023)"

-Lines 60-61: It is stated above that this is already known to be the case ("Instead, organic carbon is primary associated in the form of nanosized coprecipitates...") - further clarification of this section is needed to highlight the specific knowledge gaps.

Author's answer: Following the modification above, we rephrased the knowledge gap to clarify it: "In sum, these studies, showing various types of mineral-organic associations in different Andosols, now raise the question of their coexistence within the same soil and whether some types of associations are predominant. To determine if nanoCLICs, short-range ordered minerals with adsorbed organic matter and metal-organic complexes coexist in Andosols, or if one type is more prevalent, further nanoscale characterization of Andosol's mineral-organic associations is required."

-Line 64: "the one..." It's not likely that there is only one type of MOA susceptible to transformations under disturbance. Suggest editing to "types of MOAs" or similar.

Author's answer: We do agree the suggested change and modified to "identifying the ones vulnerable to".

-Line 66: "crop soils…" In general, the terms cultivation, cropping, crop, etc. are used interchangeably, and the set-up would benefit from some consistency and clarification of what exactly is meant by this in the context of MOA disturbance. To me, "cultivation" speaks more towards physical disturbance (tillage) while "crop/croplands" is a more general distinction. A definition of what the authors have in mind as the primary contrast/focus as it relates to MOA disturbance would help frame this more clearly.

Author's answer: As suggested by the reviewer, we removed the wording "cultivation" from the manuscript; we used instead "forest-to-crop transition" and "crop/croplands" to describes the soil converted form a forest to a cropland. The sentence appears now as follows: "In addition to characterizing existing mineral-organic associations, identifying the ones vulnerable to destruction or transformation during forest-to-crop transition is crucial for developing strategies to preserve organic matter in croplands."

-Lines 67-68: Intensified root and microbial activities after cultivation/tillage? Intensified root and microbial activities within cropping systems more generally? The direction of this effect seems specific to the nature of the cropping system. This ties into the comment earlier about setting some boundaries on the primary contrast of interest.

Author's answer: The sentence has been changed as follows: "(ii) intensified root and microbial activities within cropping systems may accelerate mineral-organic associations disruption by priming effect (Keiluweit et al., 2015; Jilling et al., 2021; Fontaine et al., 2024)".

-Lines 70-71: Do these citations (Li, Newcomb, etc.) speak to MOA destabilization in general, or with respect to cultivation/agricultural management? Please specify.

Author's answer: The sentence has been clarified as follows "However, regardless of these factors, the susceptibility of mineral-organic associations to disruption varies, depending on mineral crystallinity and binding strength (Li et al., 2017; Newcomb et al., 2017; Bernard et al., 2022)."

-Line 79: "This study aims…" From the abstract, it seems that the focus of this study is a contrast between a forested area and an adjacent cultivated agricultural field within one site (and from below, a site specifically with Fe-poor parent material). While this is probably just needing rewording here (this study isn't determining MOAs within Andosols generally, but Andosols within the specific context of the study site), the application of the findings in Andosols in general seems to pop up in other locations in the abstract (such as the last sentence of the abstract) as well as in the justifying material above. Within Andosols, there is a huge variation in mineralogy and by extension possible MOA types based on age, parent material, climate, etc., and it's important to frame the findings with more specificity (particularly to the "Fe-poor" characteristic).

Author's answer: We rephrased the sentence as follows: "This study aims to (i) investigate the predominant mineral-organic associations within an andosol (developed on Fe-poor parent material) under both forest and cropland conditions, and to explore possible differences in the types of mineral-organic associations, and (ii) to determine if mineral-organic associations in a forested Andosol developed on Fe-poor parent material (andesite) are similar to mineral-organic associations in a forested Andosol developed on Fe-rich parent material (basalt; from Jamoteau et al., 2023)."

Additionally, Andosols' variation in mineralogy, climate and ages in now mentioned in section 4.1 as follows: "Mineral-organic association in form of amorphous coprecipitates are thus found in

different Andosols: in an Andosol formed few 100 years ago on andesite parent material (Fe-poor parent material, this study), in an Andosol formed 40,000 years ago on basalt parent material (Fe-rich parent material, Jamoteau et al., 2023), and such amorphous mineral forms are likely present in many other Andosols, particularly young Andosols (e.g., Shimada et al., 2022). These Andosols show strong correlations between C and pyrophosphate-extractable metals (Alpp + Fepp), which primarily extracts metals from the least polymerized phases. However, before generalizing mineral-organic associations in the form of amorphous coprecipitates to all Andosols, future analyses must examine a broader range of Andosol types, considering various parent materials, ages and climates."

-Lines 80-82: This study doesn't seem to be comparing parent materials: is this objective primarily in contrast to prior observations (and is prior work representative of the variation within Andosols as mentioned above...?)

Author's answer: We rephrase the sentence to be specific on the comparison of an andosol under forest and cropland conditions and not comparing andosols with different parent materials. Changes appears as follows: "This study aims to (i) investigate the predominant mineral-organic associations within an andosol (developed on Fe-poor parent material) under both forest and cropland conditions, and to explore possible differences in the types of mineral-organic associations, and (ii) to determine if mineral-organic associations in a forested Andosol developed on Fe-poor parent material (andesite) are similar to mineral-organic associations in a forested Andosol developed on Fe-rich parent material (basalt; from Jamoteau et al., 2023)."

-Line 85: "in Andosols..." Andosols of one specific age, parent material, climate, etc. How can this contrast be made if you're not comparing Andosols with varying degrees of amorphous mineralogy vs. SRO phases (vs. more crystalline phases)? The framing of these hypotheses would benefit from higher specificity and ties to the actual contrasts being made in the study.

Author's answer: We have reformulated our hypotheses to be more specific to the type of andosol, as follows: "*Our working hypotheses posit that: (i) organic matter preferentially associates with amorphous mineral phases rather than short-range order minerals in relatively young Andosols (< 100 kyrs) developed on Fe-poor parent material, (ii) nanoCLICs are particularly prone to physicochemical transformations induced by cultivation, making them susceptible to destruction, and shifting the predominant mineral-organic association in the cultivated Andosol from nanoCLICs-type to adsorption of organic matter onto short-range order minerals. To probe these hypotheses, we sampled an Andosol formed on a Fe-poor parent material (andesite parent material). We conducted analyses on two Andosol topsoil that are 300 m apart, one under forest and the other subjected to three decades of cultivation.*". The age of the studied andosol is now mentioned in the materials and methods section.

-Lines 89-91: While it's reasonable to assume some degree of similarity between these locations, and justified to contrast them with respect to land-use history, there is a caveat about interpretations of "change" if the initial characteristics of the cultivated side weren't determined before cultivation began three decades ago. This is a general note as a suggestion for considerations/limitations of interpretation of study findings.

Author's answer: As mentioned by the reviewer, we are lacking data on the temporal evolution of mineral-organic associations over the past 30 years. However, the soil profile data (now presented in Figure 2) indicate consistent measurements (of C content and mineralogy) at the bottom of the soil profile, while the data above 30 cm depth vary. This can suggest, in addition to "field" criteria (distance from pits, and flat topography), that the two soil profiles can be suitable

for comparing forest-to-crop conversions. Moreover, following Reviewer#2 and Editor's comment, we acknowledge that using single soil pits for each site limits our ability to estimate spatial variation in soil properties and to make a reliable comparison between the two land uses. This limitation is now recognized in interpretation paragraphs 4.4 and 4.5. See, for example the concluding sentences of these paragraphs as follows : "Ultimately, even though the data are from only two soil profiles, if they represent a forest-to-crop conversion, our results suggest this conversion mainly affects the amount, rather than the type, of mineral-organic associations in the studied Andosol." and "Consequently, if the two analyzed Andosol profiles represent a forest-to-crop conversion, this study suggests that mineral-organic associations in the form of amorphous coprecipitates may be prone to disruption due to agricultural conversion.".

-Lines 96-98: Were these equivalent pedogenic horizons at 10-20 cm depth? Did the forest site have an organic horizon, for example, that would alter the absolute depth to a mineral horizon? Since the primary conclusions are between these single samples, the validity of the contrast needs further development. How many samples were collected for each site? It reads as one bulk sample; while sample limitation is a common situation for these high intensity imaging and characterization analyses, the sampling procedure and any replication/sample compositing relevant to interpretations needs to be clarified here.

Author's answer: We agree that the sampling method lacked details. We modified the material and method sections, as follows: "*The crop soil, converted 30 years ago, transitioned from a forest to a banana plantation followed by a cropping system with 3-year rotations of taro, sweet potatoes, yams, and fallow periods. During the crop rotation system, ploughing was carried out to a depth of 30 to 40 cm. The differences observed between these two soil profiles may primarily result from their different uses: one soil has remained in a forest for 30 years, while the other has been converted for agriculture. Soil sampling was carried out by opening soil pits and subsequently sampling 3 samples per soil horizon (on different sides of the pits), for a total sampling of ~1 kg of soil per horizon (i.e. at 0-5, 5-10, 10-20, 20-30, 30-40, 40-50, 50-60, 60-70, 70-80 cm). After sampling, the samples were kept humid at 4°C.*"

Concerning the validity of the contrast between the forest and crop andosol, we agree that the visualization of a single horizon could lead to doubt. We have now modified the bulk characterization data by showing both entire soil profiles (see Figure 2 in the revised version). Moreover, by comparing the 10-20 cm horizon for microscopy analysis, we have effectively avoided the organic horizon of forest soil (0-5 up to 5-10 cm maximum). The Figure below shows the absence of an organic horizon in the 10-20 cm horizon of the forest soil profile.

[Figure]

5 cm

10 cm

20 cm

30 cm

40 cm

50 cm

*Figure 1. Picture of the forest soil profile during sampling.*

-Lines 100-101: Following the comment above, changes relative to the baseline need to be carefully justified as a "change" without initial samples for contrast. Also, missing period between sentences.

Author's answer: We added this sentence to clarify this point: "The differences observed between these two soil profiles may primarily result from their different land uses: one soil has remained in a forest for 30 years, while the other has been converted for agriculture.".

Please refer also to Author's answer to general comment: (2) Additionally, significant changes have been made regarding the data on variations in C concentration, pH, and mineralogy between the forest and cultivated soils. Data now encompass not only the 10-20 cm horizon but the entire soil profiles (0-90 cm; 18 samples now), which greatly enhances the accuracy of the forest-to-crop conversion impact assessment. We acknowledge that using single soil pits for each site limits our ability to estimate spatial variation in soil properties and to make a reliable comparison between the two land uses. This limitation is now recognized in interpretation paragraphs 4.4 and 4.5. See, for example the concluding sentences of these paragraphs as follows : "Ultimately, even though the data are from only two soil profiles, if they represent a forest-to-crop conversion, our results suggest this conversion mainly affects the amount, rather than the type, of mineral-organic associations in the studied Andosol." and "Consequently, if the two analyzed Andosol

profiles represent a forest-to-crop conversion, this study suggests that mineral-organic associations in the form of amorphous coprecipitates may be prone to disruption due to agricultural conversion.".

-Lines 102-103: It is not immediately clear that this step is not what was used for imaging analysis. Please clarify that this process was used for bulk characterization.

Author's answer: We have clarified this point by moving the separation of fine fractions to section 2.2, dedicated to the analysis of C, pH and poorly crystalline mineral phases.

-Lines 119-122: While all sample preparation procedures have benefits and limitations, I think it's important to consider potential artifacts from the separation and preparation process that may cause the observed materials to be altered relative to the intact soil structure. For example, could there be a difference in the stability against sonication of nanoCLIC-type associations in comparison to surface adsorption of OM on SRO minerals? Could air-drying induce aggregation artifacts and lead to more clustered spatial structures? (This is especially important considering the secondary interactions between SRO phases and co-precipitates, which is super interesting!) Consideration of potential limitations or biases is needed for final interpretations of the findings by the reader.

Author's answer: We agree with the reviewer. This point has been added to the results, in part 3.2.1, as follows: "The interaction between coprecipitates and crystalline mineral phases may be secondary. While we cannot rule out the possibility that this interaction is induced by sample preparation (e.g., weak sonication followed by air-drying), the coprecipitate-mineral interaction has been found in three different mappings and even observed between mineral sheets (Figure S2), suggesting that such interactions may occur in soils."

-Lines 130-131: Why the imbalance in number of observations? This could affect overall interpretations of the degree of variability within the sample, based on more observations across a wider sample area, for example.

Author's answer: The forest soil was analyzed first, followed by the crop soil. Due to limited access to tools, fewer samples of the cultivated soil could be analyzed within the available time. However, overall results and in particular Figure 6 demonstrate that the crop coprecipitates exhibited the same heterogeneity compared to the forest coprecipitates, allowing us to approximate the same representativeness of analyses between the forest and crop andosol.

-Lines 132: What is meant here by micrometric representativeness? Please clarify.

Author's answer: the sentence has been modified as follows: "From these mappings, various zones were selected to quantify atomic proportions, ensuring micrometric representativeness of analyzed microscopy grids".

-Lines 141-145: To fully assess potential for beam damage, the relevant metric is that of electron dose (not duration, though that is also helpful information). Can electron dose (e.g. e per area per time) be calculated from available imaging parameters?

Author's answer: We agree with the reviewer's suggestion. The electron dose have been added to the manuscript, as follows: "EDX Mapping was conducted with acquisition times varying from 15 to 90 minutes (with an electron dose of $100e.\text{Å}^{-2}.s-1$).".

-Lines 169-173: With the high resolution imaging emphasis, a large number of replicate samples is not feasible, but for these bulk analyses, it's important to keep in mind that these values are

only for one sample - the range around these values is unknown and a direct quantitative contrast (e.g., 50% decrease) should be made with caution given the lack of assessment of variation in properties within the site-by-management combination. I think this information is useful for reference, but should be very clearly stated that it is a very limited sample set.

Author's answer: As mentioned above, we choose to modify Figure 1 and show data acquired on whole profiles. Section 3.1 has therefore been greatly modified, including Figure 2. Data on whole profiles greatly strengthens the results showing difference between two forest and crop topsoils. Moreover, as mentioned in our response to your general comment, the results section does not mention forest-to-crop conversion impacts but more "differences between the forest and crop soil profiles". We now mentioned having sampled only one pit in our interpretations of the differences between forest and cultivated soils in paragraph 4.4 et 4.5.

-Lines 176-179: Where is the range (50-70%) coming from? Differences in each element? There is no clear "quantitative" difference with no replication or statistical tests. Since the magnitude of the change is large, that is worth mentioning, but needs to be carefully explained in the text.

Author's answer: The sentence has been modified as follows: "The amount of Al, Fe and Si extracted by pyrophosphate was twice lower in the crop soil compared to the forest soil (on the 0-20 cm depth), indicating a lower amount of amorphous mineral phases in the crop topsoil." Moreover, as mentioned in our response to your general comment, the results section does not mention forest-to-crop conversion impacts but more "differences between the forest and crop soil profiles".

-Lines 183-186: These are interesting and relatively high magnitude changes, but presenting data in figure format and stating these values as such in the text with no assessment of variability across the site (these seem to just be one profile/point) could allow for a broader take-away than is justified by the scope of the study. Again, I think it's helpful to have these values for context for the high-resolution imaging results, but need to be more constrained in how they are stated/interpreted.
-Figure 1: Following comments in the text, the presentation of single-measurement data as such needs to be more constrained.

Author's answer: to comment on Lines 183-186 and Figure 2: Following our modifications to the manuscript and the additions of data at the profile scale, we believe we have taken into account the reviewer's comment.

-Lines 203-205: For consideration: what fundamentally defines a "co-precipitate"? To me, this implies a process, rather than a characteristic of a material - i.e. the nanoscale amorphous association that was observed. What is the lower size cutoff for a mineral phase with surface adsorption/"non-coprecipitate" properties vs. coprecipitates? For example, ferrihydrite has been described as granular structures just as small as 2 nm in size (in synthesized standards, albeit not in natural systems, see https://www.science.org/doi/full/10.1126/science.1142525?casa_token=QzpbDPaVRzsAAAAA %3A919dCIOhgBNdJLb_o2nG2zh0hH__NQdrI7z4njEFnHcVtdTWe3Kvi4_cXwGVXCHWiHBKgq8f xFangL3s), and nano-goethite forms etc. (e.g., as described in Aaron Thompson's work, e.g. https://www.sciencedirect.com/science/article/pii/S0016703710005740?casa_token=LwKJiKo K_FUAAAAA:sopD36o1OQCTo3EEXacXJ7FwRwpHtDUua4DKvNekNBreNX0NhGhxpR7aXIeVDQ HHez2b2WjFkyal). (I recognize these are both Fe phases that are less expected in this system, but just examples that come to mind). I don't disagree that the observations resemble a more homogeneous distribution that is co-precipitation processes, but I'm not sure that it is as clearly

defined as stated. More rationale here as to definition of these specific process-based forms is needed for clarity.

Author's answer: Following your comment on line 45, we modified the wording in the introduction. The term coprecipitate refers to the products of the coprecipitation process. Figure 1 now clearly shows the difference between short-range order +OM, which results from an adsorption process of OM on short-range order, and amorphous coprecipitates composed of Fe nanophases+OM, proto-imogolite+OM, nanoCLICs and metal-organic complexes. Additionally, we have clarified our final interpretations regarding the forms of amorphous coprecipitates containing Fe (see part 3.2.3). These are now attributed as follows: "Following the same logic as above, C+Al+Si-rich and Al+Si-rich zones (with less C) can be attributed to zones containing mainly coprecipitates in the form of C+Al+Si nanoCLICs and proto-imogolite+OM. Concerning the C+Fe+Al+Si rich zones, more coprecipitate forms may be present, such as C+Fe+Al+Si nanoCLICs and some Fe nanophases associated with organic matter. Therefore, all these forms of coprecipitate can be categorized as amorphous coprecipitates.". Forms of Fe nanophases associated with organic matter are now also mentioned in the summary Figure 8.

-Figure 2: The line profiles I think nicely show the increase in C signal in the amorphous phase, but I wonder about characteristics directly at the interface of the rod-like structure. The line plot implies that there is a non-linear increase in C right at the transition between phases (as indicated by the Si line profile). Were any interesting patterns found with respect to the boundary between phases? It may be partially an artifact of deposition and drying of samples, but it seems that the amorphous phases are also distributed around the edges of the rod-like mineral, which suggests there might be a secondary interaction that is important here in overall stabilization of OC in the crystalline-non-crystalline association. Note: I see this is introduced in the discussion, but more direct identification of this result is needed earlier.

Author's answer: We thank the reviewer for his comment, we added this sentence to part 3.2.1: "The interaction between coprecipitates and crystalline mineral phases may be secondary. While we cannot rule out the possibility that this interaction is induced by sample preparation (e.g., weak sonication followed by air-drying), the coprecipitate-mineral interaction has been found in three different mappings and observed even between mineral sheets (Figure S2), suggesting that such interactions may occur in soils.".

-Line 231: As mentioned above, I think of ferrihydrite phases being smaller in size, but admittedly much of this is based on synthetic studies rather than direct observations in natural soils.

Author's answer: Please refer to our response to the comment on lines 203-205.

-Lines 233: "Consisting of a few atoms…" There is a jump from the lower end of resolution (15 nm) to sizes of a few atoms as described for nanoCLICs. I think that you can definitely say that they are less than 15 nm in size, but can you distinguish between size classes below that minimum resolution?

Author's answer to comment on Lines 231,233: We agree with Reviewer 1 and we have revised the interpretation of the results section 3.2.2. The paragraph appears now as follows: "[...]. These results indicated that the mineral component is primarily composed of a mix of amorphous Al, Si, and Fe coprecipitates, even at scales down to 15 nm, and not composed of short-range ordered minerals like imogolite, allophane. The nanoscale colocalization of these elements (C, Al, Si, and Fe) demonstrates the presence of organic molecules coprecipitated with a mineral part such as

inorganic oligomers, proto-imogolite or Fe nanophases resulting from parent-andesite minerals weathering. This characterization demonstrates the wide range of coprecipitates occurring at nanoscales, such as proto-imogolite+OM, nanoCLICs, metal-organic complexes and some Fe-nanophases+OM"
Please note that in this new version of the manuscript, the atomic proportions are discussed in the following section 3.2.3.

-Figure 3: Given the focus of the paper on comparisons between forest and cultivated soil, it seems like it would contribute to the overall conclusions/interpretations to include contrasting images when they are available between the two systems in the main text.

Author's answer: We implemented the Reviewer's suggestion and split Figure 3 into two figures: one showing an example of mapping for the forest and the crop soil (Figure 5) and the other to show atomic proportions (Figure 6).

-Lines 270-272: An alternate way of looking at this is that the non-C only regions had a relatively high enrichment of 286.6 (phenolic/ketones), especially the C+Al+Fe rich area. The spectra seem to be of high enough quality that deconvolution procedures to more semi-quantitatively compare relate intensity in these regions would be a useful addition to more comprehensively compare relative abundance of C functional groups with regions of different element proportions, which I think is a really interesting element of this work.

Author's answer: We thank the reviewer for this comment. We confirm that the deconvolution of the spectrum of zone 1, rich in C+Al+Fe, clearly shows a peak at 286.6 eV (see figure below), whereas zone 5, rich in C+Al, does not show a peak at 286.6 eV. Note that in the figure below, the energy shift is not corrected, which explains why the peak at 286.6 in Figure 7 is slightly shifted in energy compared to the figure 2 below. The presence of a higher peak at 286.6 eV in the C+Fe and C+Al+Fe rich areas is now mentioned as follows :" *Moreover, compared to the area enriched in C, the speciation of C within coprecipitates richer in C+Al+Fe and C+Fe showed a higher pic at 286.6 eV attributed to organic compounds made of phenolic-C and ketonic-C.*" and explained in discussion part as follows: "*However, in areas richer in C+Al+Fe and C+Fe, a higher proportion of organic compounds made of phenolic-C and ketonic-C were observed (Fig. 7), suggesting a correlation between the presence of Fe in coprecipitates and phenolic and/or ketonic-rich compounds. This correlation could be explained by a preferential binding of phenolic compounds with Fe, highlighted in the literature (Schmidt et al., 2013; Mimmo et al., 2014).*"

**Area 1**

[Figure]

**Area 5**

[Figure]

*Figure 2. Deconvolution procedures applied to the spectra of area 1 rich in C+Fe+Al and area 5 rich in C+Al. In this figure, the energy shift is not corrected, which explains why the peak at 286.6 in figure 7 is slightly shifted in energy in this figure.*

-Figure 5: What does "C=H" represent? Please clarify what functional groups are being associated with the 285 region.

Author's answer: The associated group has been clarified (now Figure 7).

-Lines 279-383: As mentioned above, take into consideration work done to characterize amorphous organo-metallic complexes in Andosols, as well as the size variation possible within these SRO minerals (nano-sized FH, proto-imogolite, etc.) and how these might fall (or not fall!) within the concept of a "nanoCLIC". As an example: https://www.sciencedirect.com/science/article/abs/pii/S0016706122001276

Author's answer: As previously mentioned, we have significantly revised the introduction and modified the discussion to focus on the various mineral phases found in andosols. We believe

that the modifications made address the reviewer's comment, and the mentioned paper is now mentioned as follows: "Mineral-organic association in form of amorphous coprecipitates are thus found in different Andosols: in an Andosol formed few 100 years ago on andesite parent material (Fe-poor parent material, this study), in an Andosol formed 40,000 years ago on basalt parent material (Fe-rich parent material, Jamoteau et al., 2023), and such amorphous mineral forms are likely present in many other Andosols, particularly young Andosols (e.g., Shimada et al., 2022). These Andosols show strong correlations between C and pyrophosphate-extractable metals (Alpp + Fepp), which primarily extracts metals from the least polymerized phases. However, before generalizing mineral-organic associations in the form of amorphous coprecipitates to all Andosols, future analyses must examine a broader range of Andosol types, considering various parent materials, ages and climates. ".

-Lines 289-290: I agree that C is much more spatially coupled to the amorphous phases. But, there also seem to be secondary associations between the amorphous phases and the imogolite phase (as shown in Fig. 2), i.e. an association between the coprecipitated phases and the imogolite surface, as you illustrate in Fig. 6. It's not clear through the text up to Fig. 6, though, and in the set-up/abstract the potential role of interactions between SRO phases and truly amorphous phases is not thoroughly highlighted/clear. I think that is a really exciting finding and is somewhat diluted by the takeaway in the abstract and elsewhere (e.g., line 28-30).

Author's answer: We thank the reviewer for their comment. Following their advice, we have modified Figure 6, expanded the discussion on this topic in section 4.3, and mentioned this result in the abstract.

-Lines 330-331: This is a large change, but as noted above, without any replication this value should be presented with extreme caution.

Author's answer: Following the addition of data regarding bulk-scale changes between forest and cultivated soils (Fig. 2), we believe that this percentage has now been strengthened. Moreover, as mentioned in our response to your general comment, the results section does not mention forest-to-crop conversion impacts but more "differences between the forest and crop soil profiles". We now mentioned having sampled only one pit in our interpretations of the differences between forest and cultivated soils in paragraph 4.4 et 4.5.

-Line 2: Minor edit: strike "," after "Although"

Author's answer: This change has been made.

-Line 21: Throughout, I would suggest Andosol should be capitalized.

Author's answer: This change has been implemented.

-Line 31: Minor edit: Suggest editing to "cultivated"

Author's answer: This sentence has been modified.

-Line 40: Suggest editing to "loss of C"

Author's answer: This change has been implemented.

-Line 38-39: Suggest editing to "which is essential for..."

Author's answer: This change has been made.

-Line 69: Suggest editing to "which can"

Author's answer: This change has been implemented.

-Line 88: Suggest editing to "organic matter adsorption onto..."

Author's answer: This sentence has been modified.

-Line 248: Suggest editing to "further investigated"

Author's answer: This change has been implemented.

**General Comments of Anonymous referee #1**

**Manuscript quality**

The manuscript is well-written overall, and the study seems sound. The authors used microscopy and spectroscopy techniques to evaluate the co-precipitates of amorphous oligomers containing Fe, Al, and Si and organic matter in Andosols. This study focuses on understanding organomineral associations and how they are affected by land use change, especially in volcanic soils with amorphous minerals. I believe the work could be interesting for the readers of Soil, and the manuscript could be accepted after some revision.

Author's overview answer: The authors thank the reviewer for their constructive feedback, which has greatly improved the manuscript.

**Limitations on land-use comparisons**

The study builds up mainly on the detailed characterization of co-precipitates at the nanoscale, which helps us better understand how organic matter is stabilized in these soils. Perhaps the main limitation of this work is that the authors only sampled one soil profile in each land use, which makes comparisons between these lands more limited. However, once this manuscript focused more on understanding the nature of these organo-mineral associations rather than directly comparing land uses, I believe this limitation is acceptable. Yet, a few comments on this issue could be added to topic 4.4. Also some conclusions about abundance of nanoclics should be modified acknowledging these limitations (see specific comments Line 379).

Author's answer: This remark is similar to an important remark from reviewer #2. We acknowledge that the initial version of the article lacked detailed information on sampling and lacked amount of soil bulk analyses (now in the new Fig. 2). In this revised version, we have detailed the sampling method in section 2.1, which involves opening pits and taking 3 samples from different walls of the pit for each horizon. Additionally, we have included data for the entire two soil profiles in this new version (showing initially 2 samples to now 18 samples). These new results, particularly in the 0-30 cm depth, significantly strengthen the data initially presented. Additionally, following the reviewer's comments, the interpretation of mineral-organic association forms has been revised to include a diversity of amorphous coprecipitates: made of a mix nanoCLICs, proto-imogolites+OM, and some Fe nanophases+ OM and metal-organic complexes (see in the text sections 3.2 to 4.2 and in Figure 8), and not nanoCLICs only. With these changes, we believe that the statements regarding the impact of cultivation on amorphous coprecipitates (and not only nanoCLICs) are now more closely aligned with the observed results.

Moreover, we acknowledge that using single soil pits for each site limits our ability to estimate spatial variation in soil properties and to make a reliable comparison between the two land uses. The result now highlights "difference between the forest and crop soil profiles/coprecipitates" rather than "forest-to-crop conversion impacts". This limitation is now recognized in interpretation paragraphs 4.4 and 4.5. See, for example the concluding sentences of these paragraphs as follows : "*Ultimately, even though the data are from only two soil profiles, if they represent a forest-to-crop conversion, our results suggest this conversion mainly affects the amount, rather than the type, of mineral-organic associations in the studied Andosol.*" and "*Consequently, if the two analyzed Andosol profiles represent a forest-to-crop conversion, this study suggests that mineral-organic associations in the form of amorphous coprecipitates may be prone to disruption due to agricultural conversion.*".

**NanoCLICS conceptualization:**

The authors describe co-precipitates of amorphous oligomers containing Al, Si, Fe, and organic matter as nanoCLICS, as Tamrat et al. (2019) first suggested. I am confused when the authors discuss nanoclics and their comparison with short-range order minerals (SROs), as it is unclear whether nanoclics are a separate category of mineral structure or still part of SROs. The authors suggest they are a separate category for having a more amorphous and disorganized structure than assumed for SROs, as they also did in Jamoteau et al. (2023). Nonetheless, I wonder whether the observed nanoclics are simply not at the far end of the disorder spectrum of SROs. I believe this aspect is a bit ambiguous in the manuscript and the authors should state more clearly in what step we are regarding the understanding of these concepts. I believe we currently cannot say either they are or not a different type of structure, as detailed, side by side comparisons of structural attributes are missing.

Author's answer: We thank the reviewer for their comment. Based on the feedback received, significant changes have been made to the manuscript: Firstly, to avoid any misinterpretation, we clarified the nomenclature of mineral-organic association forms in the introduction. A related figure now illustrates these forms of associations (Fig. 1), greatly enhancing the manuscript's clarity. The difference between SROs+OM and nanoCLICs is now visible in Fig. 1 and is based on (i) the size of the mineral (inorganic) part of the coprecipitates, which consists of 2-3 atoms in nanoCLICs (instead of ~1000 atoms for SROs+OM), leading to an amorphous structure, and (ii) the more heterogeneous composition of the mineral (inorganic) part of nanoCLICs (containing Al, Fe, Si, and some Ca, Mg, K, etc., as per Tamrat et al., 2018 and Jamoteau et al., 2023). Secondly, regarding the observable differences between short-range order minerals and nanoCLICs, our work demonstrates that imogolite-type SROs are observed (Fig. 4 and 5), but the C was preferentially localized in more amorphous phases composed of C, Al, Si, and Fe (Fig. 5). Additionally, we have included a new demonstration (in lines 275 to 290) showing that, based on atomic proportions analyzed at the nanoscale, organic molecules can only be associated with an average of 2-3 atoms, not 1000 atoms (as in SROs) or a single atom (as in organo-metallic complexes).

Overall, the interpretation of mineral-organic association forms has been revised to include a diversity of amorphous coprecipitates: a mix of nanoCLICs, proto-imogolites+OM, and some Fe nanophase+ OM and metal-organic complexes (see text sections 3.2 to 4.2 and Fig. 8), rather than just nanoCLICs. We believe that the changes made align with the reviewer's comment.

Author's answer to specific comments:

-Materials and methods: include the type of tillage. Also, specify in which seasons the plants were cultivated and when the fallow period took place. Also specify if this system has changed or maintained consistent throughout all the 30 years from the conversion.

Author's answer: Details have been added as follows in the Materials and Methods section (including tillage depth, but frequency of tillage is unknown): "The crop soil, converted 30 years ago, transitioned from a forest to a banana plantation followed by a cropping system with 3-year rotations of taro, sweet potatoes, yams, and fallow periods. During the crop rotation system, ploughing was carried out to a depth of 30 to 40 cm.". In order to ensure a quantitative difference in C stock between the two topsoil profiles, despite the possible redistribution of C by ploughing, we have included the cumulative carbon stock data in equivalent mass in Figure 2. The data indicates that the accumulated stock over the 0-40 cm depth (maximum ploughing depth) is half

as much in cultivated soil compared to forest soil, demonstrating a difference in C stock between the forest and crop topsoils.

-Figure 1: I would change the type of graph from lines to bars/columns/boxplot. Because the lines suggest a change along time and the graph contain only two points with a straight line connecting them. This illustration suggests these changes are linear, which is not possible to know based on this comparison.

Author's answer: We agree with the reviewer and, as mentioned above, significant changes have been made regarding the data in Fig.1. The data now encompass not only the 10-20 cm horizon but the entire soil profiles (0-80 cm; 18 samples now). We have also entirely redesigned the figure (now figure 2)

-Line 347: I would modify these first two sentences because I believe it is well known that organo-mineral associations are prone to destabilization due to a variety of factors, even in buried soils as Shimada et al. 2022 evaluated.

Author's answer: We modified the beginning of this paragraph as follows: "In the literature, mineral-organic associations are considered to be stable over centennial durations (Trumbore et al., 1989; Bol et al., 2009; Feng et al., 2016; Shimada et al., 2022). However, in Andosols, which are rich in mineral-organic associations, just a few decades of cultivation can lead to the destabilization of C (Verde et al., 2005; Basile-Doelsch et al., 2009; Osher et al., 2003; Dube et al., 2009; Koga et al., 2020).".

-Line 281 – 283: I think this phrase kind of suggest that nanoclics completely replace SROs in some cases, which I think is stepping longer than the leg.

Author's answer: Following the major changes made to this manuscript, this part have been modified as follows: "Mineral-organic association in form of amorphous coprecipitates are thus found in different Andosols: in an Andosol formed few 100 years ago on andesite parent material (Fe-poor parent material, this study), in an Andosol formed 40,000 years ago on basalt parent material (Fe-rich parent material, Jamoteau et al., 2023), and such amorphous mineral forms are likely present in many other Andosols, particularly young Andosols (e.g., Shimada et al., 2022). These Andosols show strong correlations between C and pyrophosphate-extractable metals (Alpp + Fepp), which primarily extracts metals from the least polymerized phases. However, before generalizing mineral-organic associations in the form of amorphous coprecipitates to all Andosols, future analyses must examine a broader range of Andosol types, considering various parent materials, ages and climates."

Concerning the difference between amorphous coprecipitates and SROs+OM we encourage the reviewer to see the nomenclature changes made in introduction part, the new Fig.1 as well as our answers to his comment on "nanoCLICs conceptualization".

-Line 379: I am not sure whether the analyses made by the authors allow them to conclude that more than 50% of the lost C was in the form of nanoclics, because they evaluated the mineral associated organic C in the fraction below 20 microns, which can comprise different type of interactions than just nanoclics. Also I am not sure whether they can infer much about the abundance of these nanoclics because: 1) C contents in the microscopy analyses was not much different between forest and cultivated soils; 2) microscopy analyses are not the ideal tool to infer about amounts as they lack representativeness, 3) they only sampled one replicate in each land use.

Author's answer: We do agree with the reviewer and the conclusion was most entirely corrected. Following the major changes to the manuscript, the type of mineral-organic associations now comprise amorphous coprecipitates (e.i., nanoCLICs, proto-imogolite+OM and some Fe nanophases+OM and metal-organic complexes), the result now highlights "difference between the forest and crop soil profiles/coprecipitates" rather than "forest-to-crop conversion impacts", and the limitation of sampling is now recognized in interpretation paragraphs 4.4 and 4.5 The sentence initially in line 379 now appears as follows: *"While the crop topsoil was observed to have 75% less C in mineral-organic associations than the forest topsoil, alongside notable physicochemical differences, including a lower pH, the presence of similar amorphous coprecipitates in both forest and crop soils was confirmed. These differences did not seem to alter the nature of mineral-organic associations or the C content within the amorphous coprecipitates but rather suggest to affect the amount of amorphous coprecipitates in the crop topsoil."*

Author's answer: Regarding comment (1) and (2):

Exactly! Isn't it surprising? We expected to find differences in atomic proportions at the nanoscale between the coprecipitates in forest and cultivated soils (Fig. 6). If the bulk MOA-C quantities decrease but the atomic proportion of C at the nanoscale remains the same in the coprecipitates, it is very likely that the quantity of coprecipitates was lower in the cultivated soil. The elements extracted by Na-pyrophosphate also indicate a decrease in the quantity of amorphous mineral forms (which could correspond to an exaction of coprecipitates, but not only coprecipitates as explained in line 401). Moreover, we acknowledged the limitation of the areas investigated in Fig. 6, as follows in the discussion paragraph 4.4 : "*Furthermore, although the total C content in the crop topsoil was lower, the relative C proportion within the amorphous coprecipitates remained similar to the one the forest topsoil (Fig. 6, within the limits of the analyzed areas), further indicating less amount of amorphous coprecipitates in the crop topsoil (and not only a lower C content within the amorphous coprecipitates).*".

We also have been careful to use advisory words around these results, such as "indicating", "our findings suggest", "These changes did not seem to alter", "if these profiles are representative of a forest-to-crop conversion, our results suggest".

Author's answer: Regarding comment (3):

We agree with the reviewer and, as mentioned above, significant changes have been made regarding the data in Fig.1. The data now encompass not only the 10-20 cm horizon but the entire soil profiles (0-90 cm; 18 samples now). We acknowledge that using single soil pits for each site limits our ability to estimate spatial variation in soil properties and to make a reliable comparison between the two land uses. The result now highlights "difference between the forest and crop soil profiles/coprecipitates" rather than "forest-to-crop conversion impacts". This limitation is now recognized in interpretation paragraphs 4.4 and 4.5. See, for example the concluding sentences of these paragraphs as follows : "Ultimately, even though the data are from only two soil profiles, if they represent a forest-to-crop conversion, our results suggest this conversion mainly affects the amount, rather than the type, of mineral-organic associations in the studied Andosol." and "Consequently, if the two analyzed Andosol profiles represent a forest-to-crop conversion, this study suggests that mineral-organic associations in the form of amorphous coprecipitates may be prone to disruption due to agricultural conversion.".

---

## Author Response (AR2)

Author's response to reviewer's comments:

Thanks for the thorough and thoughtful response to review! I especially appreciate the additional explanation of distinctions between nanoCLICs and other phases and supporting literature in the introduction text. However, I agree with the editor's concern about the interpretation of land use conversion or change effects. The changes to wording (e.g., use of softer wording such as "may") that address the limitations of the ability to test a land-use conversion/change is appreciated, but as described in the authors' response, in the revised title, and in the manuscript, the changes do not fully address the concern as highlighted by reviewers and the editor. In my view, it is not necessarily a situation where softening the language is primarily what's needed: it is the framing of the study around forest-to-crop conversion as the main focus, given the lack of statistical power, representation of spatial variation, and direct before/after comparison of a conversion process or land-use change. The revised aims do make this more clear ("under both forest and cropland conditions"), but (1) keeping forest-to-crop conversion in the title and in some important conclusions elsewhere and (2) the revised sentences in point (ii) still imply that forest-to-crop conversion is a key focal area of the manuscript and the primary conclusions of the study.

Overall, I think additional revision of the overall framing (and a shift towards the novelty of compositional and structural insights, as suggested by the editor) is still needed. In my response below, I identified a few areas where I suggest statements need further consideration:

> Author's answer: We thank the reviewer for this thoughtful review. The requested changes have been made, including:
>
> - A change of title to focus on microscale characterization of mineral-organic associations: "*Interplay of coprecipitation and adsorption processes: deciphering amorphous mineral-organic associations under both forest and cropland conditions*".
>
> - A refinement of the objectives in the abstract specifically focusing on the identification of the types of mineral-organic associations, as follows : "Identifying the types of mineral-organic associations present within a single soil is already a known challenge and detecting those susceptible to disruption during forest-to-crop conversion is even more complex. Yet, addressing this identification challenge is essential for devising strategies to preserve organic matter in croplands. Here, we aimed to identify the predominant mineral-organic associations within an Andosol (developed on Fe-poor parent material) under both forest and cropland conditions."
>
> - Concerning comments on land-use change, we have addressed all the reviewer's points by focusing on the "differences between the forest and crop Andosol profiles" rather than the "impact of forest-to-crop conversion." Moreover, we acknowledged the limitations of comparing forest and cropland conditions in the abstract and conclusion to provide context for the study's findings (L64: "Although the sample size for comparing land-use types is limited, ..."). Additionally, we included a detailed discussion on these limitations in section 4.4 and further referenced them in section 4.5 to ensure comprehensive coverage of this critical aspect. See L420-426 "*Ultimately, these results suggest that the cropland conditions mainly affected the*

*amount, rather than the type, of mineral-organic associations in the studied Andosol. However, it is important to consider that the representativeness of a forest-to-cropland conversion in this study has some limitations: firstly, the samples come from only one soil profile taken at a single point in time, which means they may not fully capture the spatial variability within a plot. Secondly, the quantitative differences in organic carbon content and extractable pools are based on a single observation per depth. Although multiple depths within a profile contribute valuable information to the overall pattern, they do not function as replicates per se.*"

Please, find below a detailed answer to Reviewer's comments.

Lines 1-2 (Title): Keeping "forest-to-crop conversion" may over-emphasize the interpretations regarding this element of the study, and downplays the stronger microscale insights (such as secondary interactions), in my opinion.

> Author's answer: As suggested by the reviewer, we changed the title and emphasized the microscale insight and the secondary interactions. Here is the modified title: "*Interplay of coprecipitation and adsorption processes: deciphering amorphous mineral-organic associations under both forest and cropland conditions*"

Lines 32-33: While the inclusion of a depth profile does strengthen the contrast, the comparison of bulk soil characteristics is still limited by the single profile sampled per land use type. A quantitative assessment here (i.e. 75%) should be much more constrained, especially in the abstract which communicates this without the limitations of sample size being mentioned.

> Author's answer: The sentence was split, and the limitation of sample size is now mentioned in the abstract as follows: "*While the presence of similar amorphous coprecipitates in both the forest and crop Andosols was confirmed, the crop soil had 75 % less C in mineral-organic associations (in the 0-30 cm depth). Although the sample size for comparing land-use types is limited, these results suggest that the nature of mineral-organic associations remains identical despite quantitative differences.*"

Lines 35-36: In the response to review, the argument is made that the aims were revised to focus on a contrast to prior observations: "...Fe-poor parent material (andesite) are similar to mineral-organic associations in a forested Andosol developed on Fe-rich parent material (basalt; from Jamoteau et al., 2023)." This doesn't come through in the abstract, which remains focused on forest and cropland differences as the primary focus of the study. Further refinement of this focus is needed to clearly identify the takeaway from the study.

> Author's answer: The aim has been modified as follows in the abstract: "*Here, we aimed to identify the predominant mineral-organic associations within an Andosol (developed on Fe-poor parent material) under both forest and cropland conditions*". Knowing that the challenge of identifying mineral-organic associations has been clarified in the abstract ("*Identifying the types of mineral-organic associations present within a single soil is already a known challenge and detecting those susceptible to disruption during forest-to-crop conversion is even more complex. Yet, addressing this identification challenge is essential for devising strategies to preserve organic matter in croplands.*"), we choose to keep the main objective of the paper and not to detail the secondary objective in the

abstract for fluidity of reading and space constraint. All objectives are clearly detailed in the introduction.

Line 47: "Transitions from…" This is another area where the text implies the primary focus is on a forest to crop conversion/transition; revision of the framing here is needed.

Author's answer: This sentence has been modified as follows: "*In order to maintain agricultural productivity in cultivated soils, it is essential to preserve mineral-organic associations in croplands.*"

Line 60: "refuted…" don't think there is enough evidence from these observations to refute the role of SROs in general. Suggest softening the language here from "refuted" to "raised questions about the stabilizing role of short-range order minerals alone in…"

Author's answer: This sentence has been modified as follows: "*In Andosols, i.e. soils with high concentrations of mineral-organic associations, microscopy and spectroscopy analyses raised questions about the stabilizing role of short-range order minerals in the form of imogolite or allophane for C (Levard et al., 2012).*"

Lines 102-103: It should be stated here that two Andosol topsoils from single profiles were compared, with a brief explanation of why this is a useful comparison (e.g., for high-intensity imaging, limited sample sets are often necessary and enable direct evidence/visualization etc., but site-level or broader comparisons are limited).

Author's answer: This sentence has been added: "*Although only single profiles per land use type were compared (limiting broader site-level or land-use comparisons), this approach was chosen to enable high-resolution imaging and direct visualization of mineral-organic associations.*"

Lines 203-204: I think here you could state that these measurements were conducted primarily as context for the microscale measurements. This particular contrast doesn't really "probe" this process (forest-to-crop conversion) or statistically testable contrasts between forest and cropland Andosols. In general, I might also suggest keeping this information in the supplementary only and referencing it for context as needed in the discussion.

Author's answer: We agree with the reviewer and constrained our aims to show the depth profiles, as follows: "*To investigate the differences between the forest and crop soil profiles and select the appropriate horizon with quantitative differences in mineral-organic associations for micro and nanoscale mappings, we compared key physicochemical parameters between the forest and crop Andosol profiles (Fig. 2).*"

Figure 2 caption: The inclusion of the depth profiles does add a lot to the context, but it still remains single profile comparisons; as mentioned above, I suggest moving this information to the SI to use as context for observations but downplay the numerical estimation contrasts which lack any representation of error/spatial variation.

Author's answer: Following the clarification of the objective of these measurements (see response to the comment above: "… *and select the appropriate horizon with quantitative differences in mineral-organic associations for micro and nanoscale mappings*"), we

believe that showing the profile data is necessary. Additionally, these data provide context, as the reviewer mentioned.

Lines 393-394: Very interesting! I thank the authors for emphasizing this result more strongly in the abstract.

Author's answer: We thank the reviewer for their comment.

Lines 420-422: The limitation of this contrast is two-fold: first, whether or not it represents a "conversion" (since it's a space-for-time substitution) but also that the quantitative differences in OC content and extractable pools is based only on one observation per depth (multiple depths within a profile do add information to the overall pattern, but do not serve as replicates per se, and the issue of single point comparisons for a given depth or the depth profile across two horizons stands). As such, this statement is not sufficient to account for the limitation in this contrast. In general, I think it's good that the analyses covered two land use types and a range of parent materials (compared to prior studies) to get a sense of the variability in these microscale characteristics, but the overall emphasis on forest-to-crop conversion still needs further adjustment in all sections of the manuscript, including the title and abstract. To clarify, I think it's reasonable (and common, given the high time investment) to compare these two land uses with respect to microscale measurements; it's (1) the bulk measurement contrasts (e.g., 75% decrease) and (2) the argument for "conversion" (rather than differences across systems) that requires further consideration and a bit more revision in the text, in my opinion.

Author's answer: We thank the reviewer for recognizing the importance of using the contrast between forest soil and cultivated soil. Regarding paragraph 4.4, we have revised the term "conversion" at the beginning of the paragraph, as follows: "*This substantial C difference, with showed up to 75% less C in mineral-organic association in the crop topsoil compared to the forest topsoil, suggest a destabilizing effect of agricultural conversion on C in mineral-organic associations.")*.".

Moreover, we now discuss the limitations of sample representativeness in lines 420 to 424. Specifically, we have addressed the two main limitations: "*However, it is important to consider that the representativeness of a forest-to-cropland conversion in this study has some limitations: firstly, the samples come from only one soil profile taken at a single point in time, which means they may not fully capture the spatial variability within a plot. Secondly, the quantitative differences in organic carbon content and extractable pools are based on a single observation per depth. Although multiple depths within a profile contribute valuable information to the overall pattern, they do not function as replicates per se.*"

Lines 427-430: Comment above applies here as well.

Author's answer: In paragraph 4.5, we have avoided the term "forest-to-crop conversion" and instead focused on "decreases between forest and crop soil" to better reflect the study's limitations. We have also added a reference to the limitations discussed in section 4.4 to provide context for the representativeness of the samples: "*In this study, if the two analyzed Andosol profiles represent decreases between forest and crop soil (see the*

*limits of sample representativeness of forest-to-cropland conversion in 4.4), our results suggest that...".*

Minor comments:
Line 29: Revise to "i.e."
Line 30: Would "secondary" be more appropriate here? Not sure what is meant by "subsequent".
Line 60: Revise "analyzes" to "analyses"
Line 67: Suggest revising to "These results suggest that in some situations"
Line 76: Suggest revising to "mineral-organic associations in Andosols"
Line 93: Suggest revising to "Andosol"

Author's answer: we thank the reviewers for their minor comments. All suggested changes have been implemented.